

# Optical receiver characterisations and corrections for ground-based and airborne measurements of spectral actinic flux densities

Birger Bohn[1] and Insa Lohse[1,2]

[1]Institut für Energie- und Klimaforschung, IEK-8: Troposphäre, Forschungszentrum Jülich GmbH, 52428 Jülich, Germany
[2]Deutscher Wetterdienst, Bildungszentrum Langen, 63225 Langen, Germany

**Correspondence:** B. Bohn (b.bohn@fz-juelich.de)

**Abstract.** Solar actinic radiation in the UV/VIS range perpetuates atmospheric photochemistry by inducing photolysis processes which form reactive radical species. Photolysis frequencies are rate constants that quantify the rates of photolysis reactions and therefore constitute important parameters for quantitative analyses. Photolysis frequencies are usually calculated from modelled or measured solar spectral actinic flux densities. Suitable measurements techniques are available but measure-

ment accuracy can suffer from non-ideal $2\pi$ or $4\pi$ solid angle reception characteristics of the usually employed $2\pi$ optical receivers, or receiver combinations. These imperfections, i.e. deviations from an angle-independent response, should be compensated by corrections of the measured data. In this work, the relative angular sensitivities of four commonly used $2\pi$ quartz receivers were determined in the laboratory in a range 280 – 660 nm. Based on this information, the influence of the non-ideal responses on measured spectral actinic flux densities for ground-based and airborne applications was investigated for a wide

range of atmospheric conditions. Spectral radiance distributions and contributions of direct, diffuse downward and diffuse upward spectral actinic flux densities were calculated with a radiative transfer model to derive the corrections. The intention was to determine the ranges of possible corrections under realistic measurement conditions and to derive simple parametrizations with reasonable uncertainties. For ground-based $2\pi$ measurements of downward spectral actinic flux densities, corrections typically range $<10\%$ dependent on wavelength and solar zenith angle, with $2 – 8\%$ uncertainties covering all atmospheric

conditions. Corrections for $4\pi$ airborne measurements were determined for the platforms Zeppelin NT (New Technology) and HALO (High Altitude and Long Range Research Aircraft) in altitude ranges 0.05 – 2 km and 0.2 –15 km, respectively. Total, downward and upward spectral actinic flux densities were treated separately. In addition to various atmospheric conditions, different ground albedos and small ($<5°$) aircraft attitude variations were considered in the uncertainties, as well as aircraft headings with respect to the sun in the case of HALO. Corrections for total and downward spectral actinic flux densities again

typically range $<10\%$ dependent on wavelength, solar zenith angle and altitude, with $2 – 10\%$ uncertainties covering all atmospheric conditions for solar zenith angles below 80°. For upward spectral actinic flux densities corrections were more variable and significantly greater, up to about $-50\%$ at low altitudes and low ground albedos. A parametrization for corrections and uncertainties was derived using uncorrected ratios of upward/downward spectral actinic flux densities as input, applicable independent of atmospheric conditions for a given wavelength, solar zenith angle and altitude. The use was limited to conditions

with solar zenith angles $<80°$ when direct sun radiation cannot strike upward and downward looking receivers simultaneously. Examples of research flights with the Zeppelin and HALO are discussed, as well as other approaches described in the literature.





# 1 Introduction

Photodissociation of atmospheric gas-phase constituents by solar UV/VIS radiation is essentially influencing atmospheric chemistry and composition through the formation of highly reactive photo-products. These intermediates, or secondary products like OH, can initiate oxidizing chain reactions and lead to other reactive species like $O_3$. The rates of photolysis processes are quantified by first-order rate constants denoted as photolysis frequencies which are important parameters because they directly or indirectly determine the lifetimes of many atmospheric species. Accurate knowledge is therefore essential for a quantitative understanding of atmospheric photochemistry. Photolysis frequencies can be determined from solar spectral actinic flux densities $F_\lambda$. For example, $j(NO_2)$, the rate constant of the process $NO_2 + h\nu(\lambda \leq 420\ \text{nm}) \longrightarrow NO + O(^3P)$, is calculated by integration over the relevant wavelength range:

$$j(\text{NO}_2) = \int F_\lambda(\lambda) \times \sigma_{\text{NO}_2}(\lambda) \times \phi_{\text{O}(^3\text{P})}(\lambda)\,\text{d}\lambda \tag{1}$$

$\sigma_{\text{NO}_2}$ and $\phi_{\text{O}(^3\text{P})}$ are the absorption cross sections of $NO_2$ and the quantum yields of the photo-product $O(^3P)$, respectively. $F_\lambda$ is inserted in molecular units ($\text{cm}^{-2}\text{s}^{-1}\text{nm}^{-1}$). Photolysis frequencies of other photolysis processes can be calculated accordingly by inserting the respective parameters of the precursor molecules. Spectroradiometry, a technique to measure $F_\lambda$ in the relevant UV/VIS spectral range is therefore the most convenient experimental method to determine photolysis frequencies. Measurements of $F_\lambda$ are important for many field studies mainly because the strong and variable influence of clouds on actinic radiation is hard to predict by radiative transfer models unless detailed local cloud information is available. A general overview of techniques to derive photolysis frequencies in the atmosphere by radiometric and chemical methods, as well as by radiative transfer models is given by Hofzumahaus (2006).

The radiometric determination of $F_\lambda$ in the atmosphere is complicated by two experimental challenges related with (i) the accuracy of measurements in the UV-B range and (ii) the quality of optical receivers for actinic radiation. For aircraft measurements these issues are particularly relevant:

(i) UV-B radiation is strongly diminished in the lower atmosphere by stratospheric ozone but highly important for tropospheric ozone photolysis and OH formation. Aircraft deployments require both, high time resolution and high UV sensitivity which can be achieved by CCD array spectroradiometers. However, because these instruments are single-monochromator based, the weak UV-B range is significantly affected by stray light, i.e. by radiation that is non-regularly reflected inside monochromators. Instrument calibrations and field data analyses therefore require special procedures to minimize the stray light influence. In a previous study a suitable approach was described for a widely used type of spectroradiometers (Bohn and Lohse, 2017).

(ii) Spectral actinic flux density $F_\lambda$ is obtained upon integrating the directional quantity spectral radiance $L_\lambda$ over all solid angles $\omega$:

$$F_\lambda(\lambda) = \int\limits_0^{4\pi} L_\lambda(\omega, \lambda)\text{d}\omega \tag{2}$$



In contrast to spectral irradiance, no polar angle dependent weighting of $L_\lambda$ is applied and no sign distinction between upward and downward flux densities because from the perspective of gas-phase molecules radiation is received with the same efficiency regardless of the direction of incidence. Therefore, the ideal optical receiver for actinic radiation has an angle-independent reception sensitivity and a $4\pi$ solid angle field-of-view. A corresponding $4\pi$ optical receiver (Teflon sphere) with adequate properties was described in the literature (Eckstein et al., 2003). However, technically $2\pi$ receivers covering a hemisphere are more practicable and often sufficient, e.g. for many ground-based applications under conditions with low ground albedo. On the other hand, owing to the greater importance of upward radiation, airborne measurements require $4\pi$ reception characteristics which is accomplished by two $2\pi$ receivers on the top and bottom fuselage of the aircraft. Because the usually employed quartz-dome receivers have vertical extensions and adequate horizontal shielding can be difficult for technical reasons (Sect. 2), some cross talk to the opposite hemisphere is typical. Receiver specific corrections are therefore necessary to compensate for cross-talk as well as for other imperfections. Corresponding corrections were derived in the literature for ground-based and airborne applications (Volz-Thomas et al., 1996; Shetter and Müller, 1999; Hofzumahaus et al., 1999; Hofzumahaus et al., 2002; Eckstein et al., 2003; Jäkel et al., 2005; Stark et al., 2007; Bohn et al., 2008). These corrections were based on laboratory measurements of angular sensitivities of the receiver optics and radiative transfer calculations of spectral actinic flux density contributions from direct, diffuse downward and diffuse upward radiation. However, except for the studies by Volz-Thomas et al. (1996) and Jäkel et al. (2005), estimated mean corrections and uncertainties were applied, independent of actual measurement conditions.

In this work, an extended approach was developed by consulting spectral radiance distributions from radiative transfer calculations for a wide range of atmospheric conditions. Corrections were derived as a function of wavelength, altitude and solar zenith angle for two pairs of receiver optics that were deployed during several missions on the airborne platforms Zeppelin NT (New Technology) and HALO (High Altitude and Long Range Research Aircraft, Gulfstream G550), as well as for ground-based pre- and post-flight comparisons of downward spectral actinic flux densities. The objective was to determine as possible accurate corrections with realistic uncertainty estimates and to derive parametrizations that are easily applicable under all measurement conditions. The uncertainties of the corrections add to those from the radiometric calibrations which are typically small and range around 5-6% based on traceable spectral irradiance standards (Bohn and Lohse, 2017). Consequently, even small corrections and small improvements of uncertainties are significant.

## 2 Actinic receiver optics and installations

The employed $2\pi$ actinic receiver optics were developed by Meteorologie Consult GmbH based on an original design by Junkermann et al. (1989) with modifications implemented by Volz-Thomas et al. (1996) and have been widely used in atmospheric research since more than 25 years (Volz-Thomas et al., 1996; Shetter and Müller, 1999; Hofzumahaus et al., 1999). The receivers are composed of a stack of sandblasted, elongate quartz domes covering a quartz rod in an aluminium housing. The plain outer dome surface is sealed against a black-anodized aluminium base flange (Fig. 1). Radiation that enters the receiver is multiply scattered and partly transmitted by the quartz domes until it reaches a sandblasted surface at the bottom of the





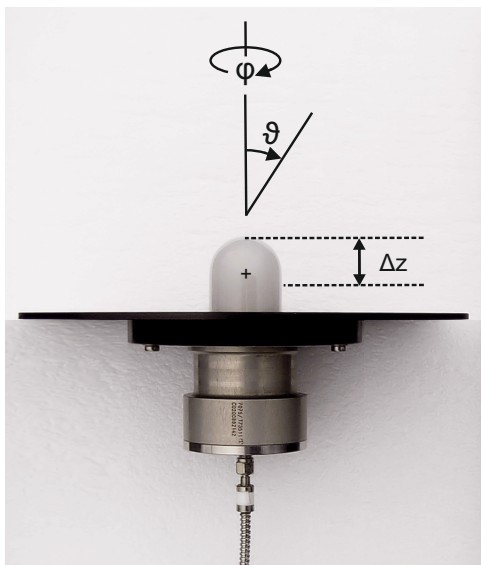

**Figure 1.** Photograph of a $2\pi$ actinic radiation receiver with quartz dome (top) and optical fiber connection (bottom). Polar and azimuth angles of incidence $\vartheta$ and $\varphi$ are indicated. Ideally the receiver collects radiation from a hemisphere ($\vartheta \leq 90°$). A typical distance $\Delta z$ of the equivalent plane with respect to the quartz dome tip is indicated by the dashed lines (Sect. 3.1). The central cross indicates the normal position of the rotational axis for $\vartheta$ dependent measurements (Sect. 3.1). The optical fiber connected at the bottom guides transmitted radiation to a spectroradiometer. The receiver housing was designed for HALO. In this photograph it is equipped with a lighter, 200 mm round top flange substitute for ground and laboratory measurements, see Fig. 2 for comparison.

quartz rod. This surface forms a virtual light source that can be captured by an optical fiber, eventually guiding the radiation to a spectroradiometer or other detectors. The distances of the domes from each other can be adjusted for optimum angular response of the receiver, i.e. an ideally angle-independent sensitivity within a hemisphere. However, despite adjustments some receiver-specific imperfections typically remain. In particular the vertical extension that is necessary for sufficient sensitivity at

near-horizontal incidence, can cause cross-talk to the other hemisphere which is significant for aircraft measurements because of typically high spectral radiances in both hemispheres. The cross-talk can be reduced by fitting the receiver base flanges into larger, black-anodized or varnished flanges, or by using horizontal shadow rings that act as artificial horizons.

Ground-based installations in this work were occasionally set up on a roof platform at Forschungszentrum Jülich for the purpose of comparisons with a reference instrument before and after airborne deployments (Bohn and Lohse, 2017). Ground-

based measurements were confined to downward actinic flux densities with aircraft top and bottom receivers facing the upper hemisphere using the original aircraft flanges or matching substitutes. Because the local surroundings had a low ground albedo (roofing felt), cross-talk effects were insignificant for this setup as during previous intercomparisons (Bohn et al., 2008).

Aircraft installations of the receivers were adapted to the specific requirements of the Zeppelin and HALO. For the Zeppelin the top-receiver covering the upper hemisphere was installed on the roof cover of a rectangular instrument box that was sitting

on top of the airship envelope. An about 1 m$^2$ wide roof area surrounding the receiver flange was covered with black matted





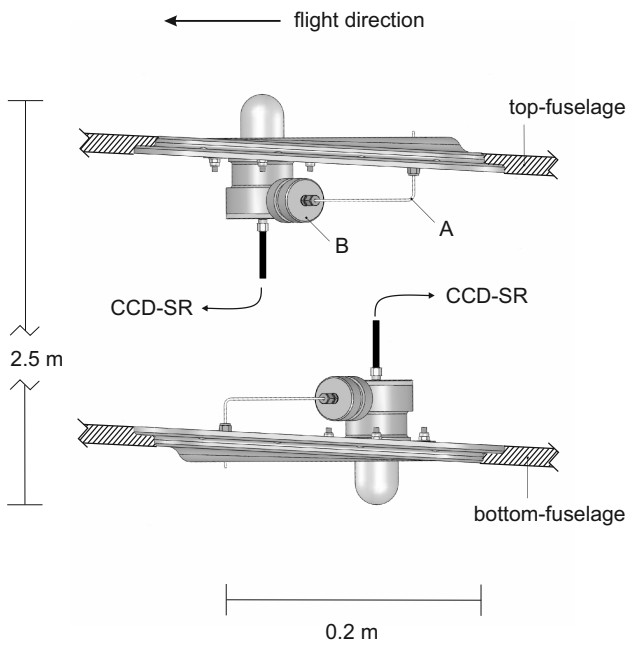

**Figure 2.** Scheme of the $4\pi$ actinic radiation setup on HALO, composed of two $2\pi$ receivers. The tilts in the instrument flanges compensate for a typical in-flight pitch angle of HALO. Receiver housings are pressure balanced via 1/16" capillaries (A) connected through cartridges with a drying agent mounted at the housing sides (B).

foil, resulting in an effective horizontal shielding. The reflective properties of the plastic foil were investigated in the laboratory (Sect. 3.2). The bottom receiver covering the lower hemisphere was mounted under the cabin in an extension flange to avoid shadings by other inlets. In this case the field-of-view was limited by the 200 mm receiver flange alone, unaffected by any airship structure. A scheme of the setup is shown in Fig. S1 of the Supplement.

For the HALO aircraft aerodynamic requirements were more demanding and receivers were built into robust instrument plates compatible with the aircraft notches ($\approx$ 200 mm $\times$ 300 mm). The same construction was used for top and bottom receivers, but to compensate for the typical pitch angle of HALO under normal flight conditions, instrument plates were slightly tilted by 3.3° in opposite directions on the top and bottom fuselage in the middle-front section of the aircraft. This setup is shown schematically in Fig. 2 and was repeatedly employed for two specific inlet configurations named FLT and FLV

in the following. In a third configuration denoted FLN, the bottom receiver was placed in the rear section of the aircraft. The ascending slope of the bottom fuselage in the rear section was compensated by turning the instrument plate by 180°, again resulting in horizontal orientations under normal flight conditions. Another factor was the glossy white paint of HALO that caused specular reflections striking the receivers in a narrow range of incident angles. Laboratory measurements were made to estimate the influence of these reflections which affected the configurations FLT and FLN (Sect. 3.2). In the FLV configuration

the instrument plates were built into larger ($\approx$ 60 cm) black-anodized flanges that effectively prevented the influence of aircraft reflections but had no effect on the field-of-view because they were shaped as the aircraft fuselage. The horizontal shielding





by the aircraft fuselage on average was around 6° below the horizon but different in lateral and parallel directions (Sect. 3.2). The use of larger, flat flanges to improve the horizontal shielding of the receivers was not feasible for this comparatively small aircraft without expensive flight tests. Moreover, an attempt by the manufacturer to downsize the receivers to minimize
cross-talk effects without degrading the $2\pi$ reception characteristics was not successful in the run-up of the HALO integration.

For field and laboratory measurements, receiver optics were connected with CCD array spectroradiometers (CCD-SR) with optical quartz fibers of suitable lengths (2–12 m). The CCD-SR were developed by Meteorologie Consult GmbH for atmospheric measurements of spectral actinic flux densities. The instruments are composed of a single monochromator (Carl Zeiss, MCS-CCD) with a spherical refraction grating and a temperature stabilized CCD array detector (Hamamatsu, S7031-0906S).
These components were built into compact aluminium housings that were placed in 19 inch flight-rack mounts. Actinic flux density spectra were measured with a spectral resolution of about 2 nm in a wavelength range 280–650 nm with a time resolution of 1–3 s dependent on the aircraft. More details on the employed CCD-SR, the calibration procedure and the data analysis can be found in a previous paper (Bohn and Lohse, 2017). The CCD-SR were also used for the laboratory characterizations of the optical receivers utilizing extended integration times of up to 1 s and repeated measurements (10–100) to improve signal-to-
noise ratios in the UV range (Bohn and Lohse, 2017). However, it should be noted that the targeted receiver-specific properties and the resultant corrections are independent of the radiometric detection method.

## 3    Angular sensitivities

### 3.1    $2\pi$ receivers

The knowledge of the relative angular sensitivities of the optical receivers is the basis to assess the uncertainties and to correct
atmospheric measurements of spectral actinic flux densities. Angle dependent sensitivity measurements were carried out in the laboratory with a goniometric setup on an optical bench where the receivers including their aircraft flanges were positioned at different incident angles relative to a stabilized point light source (1000 W tungsten halogen lamp). Polar angles of incidence $\vartheta$ were defined here as usual in geometric optics and indicated in Fig. 1 for a $2\pi$ receiver. Azimuth angles $\varphi = 0°$ refer to fixed positions on the receiver base flanges which correspond to the flight directions of the aircraft-installed receivers. Pictures of
the goniometric setup are shown in Fig. S2 of the Supplement.

Angle dependent measurements of lamp spectra were made in a range for $0° \leq \vartheta \leq 115°$. By extending the range beyond 90°, the crosstalk for each receiver was investigated, including the shading effects of the aircraft specific flanges. Azimuth angles were changed in 45°-steps in a range $0° \leq \varphi \leq 360°$.

Following the notations introduced by Hofzumahaus et al. (1999), relative angular sensitivities $Z_\mathrm{p}$ were determined by
normalizing background corrected signal spectra $S$ with those obtained at normal incidence ($\vartheta = 0°$):

$$Z_\mathrm{p}(\lambda, \vartheta, \varphi) = \frac{S(\lambda, \vartheta, \varphi)}{S(\lambda, \vartheta = 0, \varphi)}. \tag{3}$$

For an ideal receiver $Z_\mathrm{p} = 1$ for all wavelengths at $\vartheta \leq 90°$ and $Z_\mathrm{p} = 0$ for $\vartheta > 90°$.





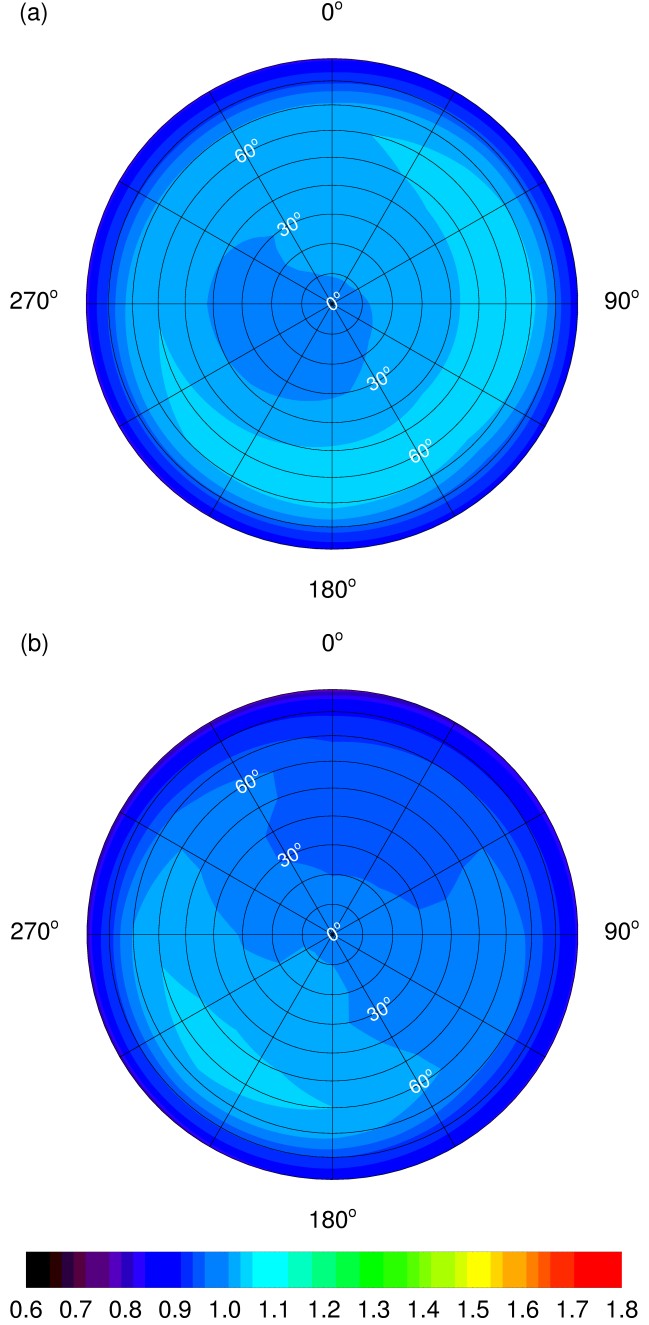

**Figure 3.** Contour plots of hemispherical relative angular sensitivities $Z_\mathrm{p}$ of HALO $2\pi$ (a) top and (b) bottom receivers at 400 nm (top views). Azimuth angles of 0° correspond to flight directions of aircraft-installed receivers. Polar angles of incidence are indicated (white). Note that cross-talk to the lower hemisphere is invisible in this representation. The color scale was chosen for better comparability with Fig. 5.





The index of $Z_\mathrm{p}$ indicates the use of a point light source in front of which the receiver was rotated. For a point light source the problem is that the flux density strongly depends on distance following an inverse square law. As a consequence, for actinic

radiation receivers with vertical extensions, the concept of an equivalent plane receiver is used for calibrations with irradiance standard lamps: the lamp position is adjusted for a receiver-specific distance $\Delta z$ with respect to the quartz dome tip. Typical $\Delta z$ range around 20 mm for an incident angle $\vartheta = 0°$ as indicated in Fig. 1. They have to be determined experimentally for each receiver to ensure accurate calibrations (Hofzumahaus et al., 1999; Bohn and Lohse, 2017). In this work, $\Delta z$ values were also determined for $\vartheta = 90°$ which turned out to be smaller by 8–15 mm. The polar angle dependent differences correspond to

small but significant signal changes that can affect the angle dependent $Z_\mathrm{p}$ measurements at the lamp distances used. Enhanced distances $z$ between lamp and receiver would be favourable to avoid this problem but greater distances also result in smaller signals, dependent on lamp power, wavelength and the detector used.

To avoid uncertainties caused by the potentially $\vartheta$-dependent $\Delta z$, the laboratory procedure was revised. Angle dependent measurements were performed at two lamp distances of $z = 400$ mm and $z = 800$ mm with respect to the equivalent plane at

$\vartheta = 0$. The final $Z_\mathrm{p}$ were then determined by a two-point extrapolation towards an inverse distance of zero, i.e. they correspond to a hypothetical infinite distance $z$. The influence of distance on the measured $Z_\mathrm{p}$ was generally small but not negligible at least for two of the employed receivers. More details on the experimental approach and a formal derivation of the two-point-method are given in Sect. S2.1 of the Supplement.

Contour plots of the finally derived $Z_\mathrm{p}$ are shown in Fig. 3 for the HALO top and bottom receivers for a wavelength of

400 nm as an example. Corresponding plots for the Zeppelin receivers are shown in Fig. S4 of the Supplement. An azimuthal equal-area projection was chosen to correctly reproduce the solid angle contributions for different polar angles relevant for actinic flux density measurements. Because of the rotational symmetry of the receivers, dependencies on azimuth angles are minor. Cross talk effects are obviously invisible in Fig. 3. Similar plots for the opposite hemispheres are not shown because the values are mostly zero except for narrow $\leq 15°$ bands close to the horizon. Instead, Fig. 4 shows azimuthal mean $Z_\mathrm{p}$ values for

the HALO top and bottom receivers for selected wavelengths where the crosstalk to the other hemisphere becomes visible. This cross-talk quickly diminishes above 90° and vanishes at around 105°. The $Z_\mathrm{p}$ dependencies on polar angle and the wavelength dependence are slightly different for the different receivers. The properties of the $2\pi$ receivers investigated here are similar to those shown in previous work using the same type of receivers (Shetter and Müller, 1999; Hofzumahaus et al., 1999; Jäkel et al., 2005; Bohn et al., 2008). Corresponding plots for the Zeppelin receivers are shown in Fig. S5 of the Supplement.

## 3.2  $4\pi$ aircraft assemblies

For ground-based measurements, the $Z_\mathrm{p}$ data of Figs. 3 and 4 are directly applicable for the calculation of correction functions (Sect. 5). On the other hand, for airborne measurements the combined total sensitivities of the receivers installed on the top and bottom fuselage have to be considered. As an example, Fig. 5 shows contour plots of total relative angular sensitivities $Z_\mathrm{p}^\mathrm{T}$ of the FLT configuration on HALO in the upper and the lower hemisphere for a wavelength of 400 nm. The $Z_\mathrm{p}^\mathrm{T}$ comprise

the combined effects of the $Z_\mathrm{p}$ of top and bottom receivers, geometrical restrictions of the fields of view by the aircraft, and fuselage reflections. More details on field-of-view effects and fuselage reflections are given in Sect. S2.2 and S2.3 of the



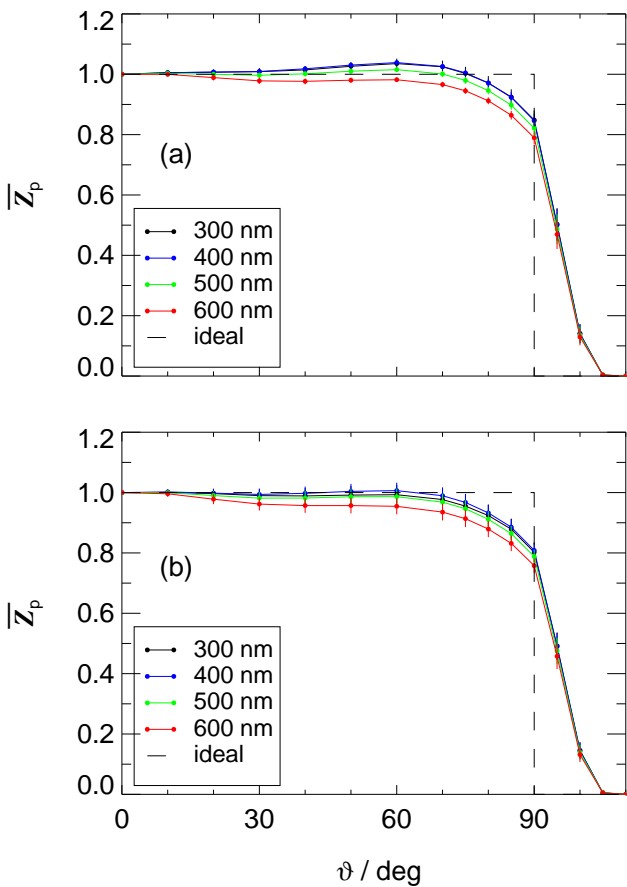

**Figure 4.** Azimuthal averages of relative angular sensitivities $Z_p$ of HALO (a) top and (b) bottom receivers for selected wavelengths. Error bars indicate standard deviations of the azimuthal variabilities. The sensitivity of an ideal $2\pi$ receiver is shown for comparison (dashed line). The receivers were built into substitutes of aircraft flanges as shown in Fig. 1.

Supplement. The range of incidence angles in Fig. 5 was extended to 0–180° with $\vartheta = 0°$ and 180° corresponding to zenith and nadir directions, respectively. The cross talk effects on $Z_p^T$ are most pronounced towards the aircraft sides where the field-of-view restrictions were smallest because of the curved fuselage. Towards the flight direction the cross talk is correspondingly

smaller and also influenced by the 3.3° tilt angle of the aircraft (Fig. 2). In the rear direction, the field-of-view in the lower hemisphere was for this configuration restricted by a containment on the bottom fuselage. This restriction prevented cross talk to the upper hemisphere in a rearward section visible in panel (a) and causes the dark area close to the horizon in panel (b) where radiation was blocked. For $\vartheta < 80°$ and $\vartheta > 100°$ the $Z_p^T$ correspond to those shown in Fig. 3. Similar plots for the two other HALO configurations FLN and FLV as well as for the Zeppelin are shown in Figs. S8, S10 and S12 of the Supplement.

Azimuthal averages of the data in Fig. 5 are plotted in panel (a) of Fig. 6. In this representation the contributions of the top receiver $Z_p^Z$ (zenith-oriented) and bottom receiver $Z_p^N$ (nadir-oriented) become visible. At $\vartheta > 80°$ total sensitivities are





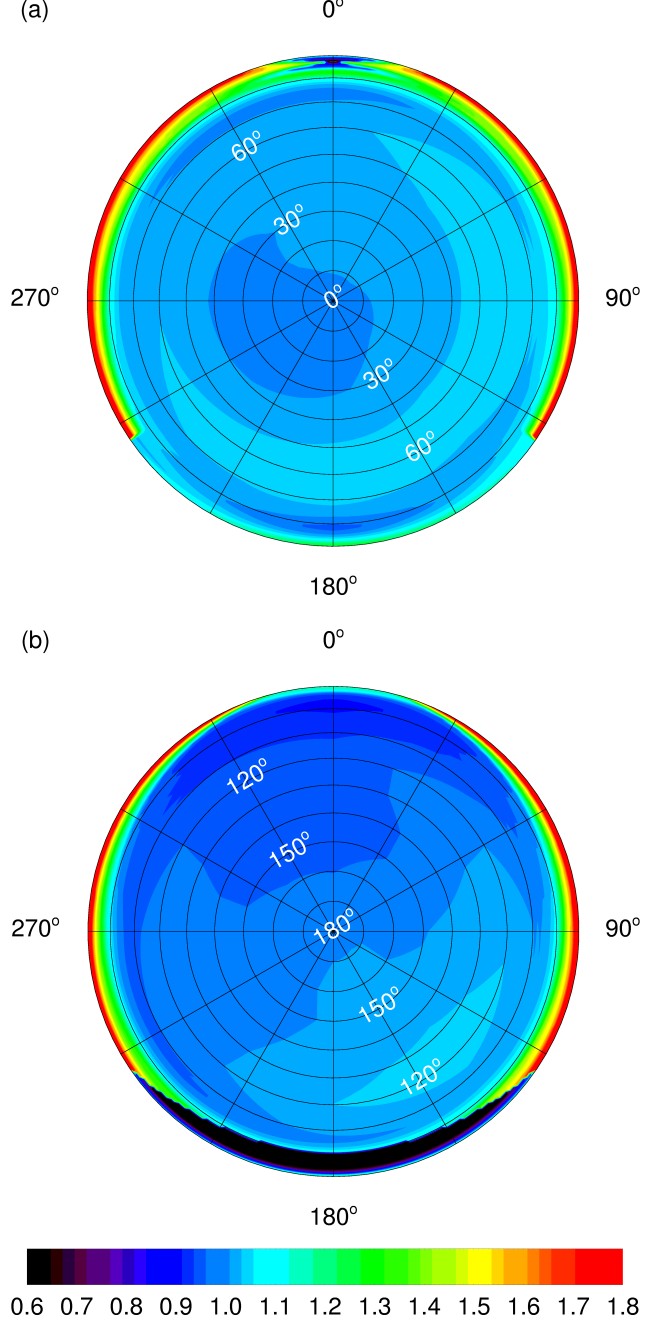

**Figure 5.** Contour plots of HALO total relative angular sensitivities $Z_\mathrm{P}^\mathrm{T}$ of the FLT $4\pi$ receiver combination at 400 nm (top views). (a) Upper hemisphere, (b) lower hemisphere. An azimuth angle of 0° corresponds to the flight direction. Polar angles of incidence are indicated (white). For the FLT configuration field-of-view and fuselage reflection effects are considered including the influence of a containment on the lower fuselage causing missing cross-talk in panel (a) and dark areas in panel (b) in rearward directions. Note that compared to Fig. 3, the features in the lower panel are laterally reversed because the receiver is now facing downwards.



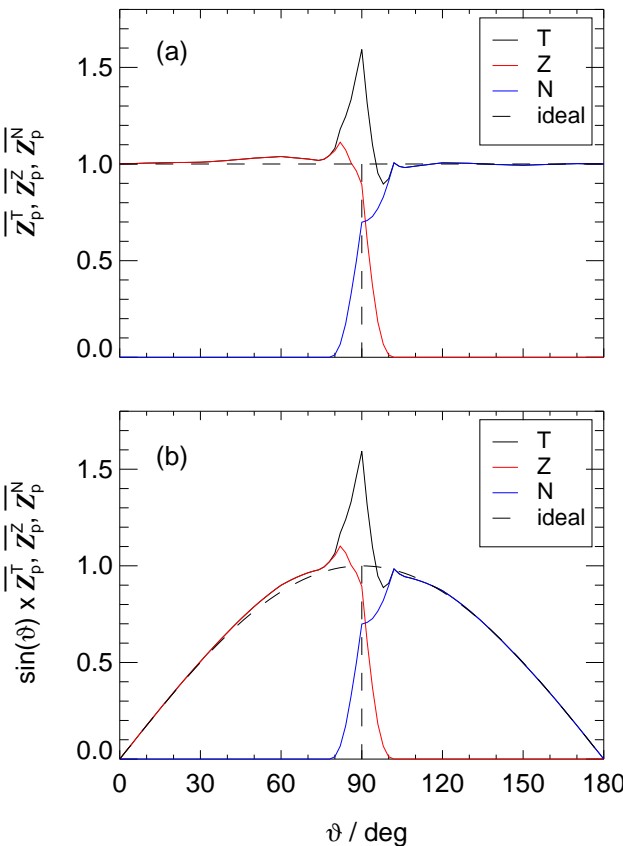

**Figure 6.** (a) Azimuthal averages of total relative angular sensitivities $Z_{\mathrm{p}}^{\mathrm{T}}$ (T) shown in Fig. 5 with contributions $Z_{\mathrm{p}}^{\mathrm{Z}}$ (Z) and $Z_{\mathrm{p}}^{\mathrm{N}}$ (N) of top and bottom receivers, respectively, for a wavelength of 400 nm. The sensitivities of ideal $2\pi$- and $4\pi$-receivers are shown for comparison (dashed lines). (b) The same data as in (a) but multiplied with $\sin(\vartheta)$ to account for the $\vartheta$-dependence of solid angle contributions.

enhanced (on average) by up to a factor of about 1.6 at $\vartheta = 90°$ because radiation can strike both receivers simultaneously caused by the non-ideal field-of-view limitations. As a consequence, radiance contributions from polar angles around 90° have to be corrected substantially which also applies to direct sun actinic flux densities at low sun.

In panel (b) of Fig. 6 relative sensitivities were multiplied with $\sin(\vartheta)$ to account for the solid angle contributions consistent with the $\vartheta$-dependent areas in the projections of Figs. 3 and 5. In the simplest case of an isotropic radiance distribution, the data shown in panel (b) of Fig. 6 would lead to an overestimation of measured actinic flux densities that correspond to the integral of the $\sin(\vartheta) \times Z_{\mathrm{p}}^{\mathrm{T}}$ curve divided by the integral of the ideal $\sin(\vartheta)$ curve. In this example, the ratio is 1.045 which is suitable to correct measurements at 400 nm, albeit under the special conditions of constant radiances. In order to obtain more

realistic corrections, wavelength dependent direct sun contributions and diffuse spectral radiance distributions are required. This information is usually not available under measurement conditions. Correction functions were therefore calculated based on results from a radiative transfer model.


## 4 Radiative transfer calculations

### 4.1 Model settings

Distributions of diffuse spectral radiances were calculated with the radiative transfer model uvspec from the libRadtran package (version 2.0.4) (Mayer and Kylling, 2005; Emde et al., 2016). The purpose was not to obtain radiance distributions for actual measurement conditions. Rather a range of atmospheric scenarios was created that should ideally cover all realistic measurement conditions. Main model input parameters are listed in Tab. 1. The radiative transfer equation solver DISORT in pseudo-spherical geometry was utilized (Buras et al., 2011) with 16 streams to obtain accurate spectral radiance output

suitable to calculate spectral actinic flux densities by numerical integrations (Kylling et al., 1995; Hofzumahaus et al., 2002). Calculations were made for 12 different solar zenith angles and an arbitrary solar azimuth angle of 180°. The radiance output was generated with a step size of 2° in 0–180° ranges for polar and azimuth angles of incidence, resulting in 8280 spectral radiance values for each wavelength. In subsequent calculations, radiances in the azimuth range 180–360° were produced by inversion of the 0–180° results. In addition, spectral actinic flux densities for total downward, diffuse downward and diffuse

upward radiation were calculated for consistency checks and as an additional input for the evaluation of correction functions (Sect. 5).

All model calculations were made in the wavelength range 290–660 nm using 5 nm steps below 310 nm and 20 nm steps above 320 nm, i.e. the total number of wavelengths was confined to 23. This is justified because, except for the UV-B range which is affected by stratospheric ozone, a smooth change of radiance distributions with wavelength was expected. Despite

this coarse wavelength sampling, a triangular response function with a FWHM of 1.7 nm was adopted in the model to allow for an optional comparison of the model output with measurements (Bohn and Lohse, 2017).

### 4.2 Atmospheric scenarios and variables

A number of atmospheric scenarios was devised to simulate realistic measurement conditions. An atmospheric scenario was defined by a cloud case, a ground albedo case and an aerosol case. For each scenario, calculations were made for up to 11

altitudes (Tab. 1). The total ozone column was fixed at a typical value of 300 DU for the majority of the model calculations. For selected altitudes of 1 km and 10 km, additional calculations were made for 200 DU and 400 DU to examine the influence of ozone columns. The ground elevation was set to mean sea level except for additional clear-sky calculations at a ground elevation of 1 km and heights above ground of 0 km and 1 km.

Four cloud cases were distinguished: (i) clear-sky, no clouds (Cl), (ii) an optically thin, high-level cirrostratus layer (Cs),

(iii) an optically thick medium-level altostratus layer (As) and (iv) an optically thick low-level stratus layer (St). In the model, clouds were idealized as homogeneous layers. The idea was to reproduce conditions with HALO flying below, within or above clouds at different altitudes and the Zeppelin always flying below any clouds. Cloud micro- and macro-physical properties, as well as cloud optical depths (COD) are listed in Tab. 2. More details on the implementation of clouds in the model are given in Sect. S3.1 of the Supplement.



**Table 1.** Input parameters of the radiative transfer model libRadtran for the calculation of atmospheric spectral radiance distributions and spectral actinic flux densities (total downward, diffuse downward and diffuse upward). More details are given in Sect. 4 and in Sect. S3 of the Supplement.

| Main model parameters | |
|---|---|
| Extraterrestrial spectral irradiance | Atlas plus Modtran |
| Atmospheric profiles | US standard atmosphere |
| Wavelength range | 290–660 nm |
| Ozone column | 300 DU[a] |
| Aerosol | Default[b] |
| Ground elevation | Mean sea level[c] |
| Ground pressure | 1013 hPa |
| Spectral ground albedo | Vegetation (mean)[d], snow, water |
| **Varied model parameters** | |
| Cloud cases[e] | Clear-sky (Cl), cirrostratus (Cs), altostratus (As), stratus (St) |
| Altitude (km) | 0.00, 0.05, 0.1, 0.2, 0.5, 1.0, 2.0, (3.5)[f], 5.0, 10, (11)[f], 15 |
| Spectral ground albedo (470 nm)[g] | 0.02, 0.04, 0.07, 0.80 (snow), $\approx$0.03 (water) |
| Aerosol optical depth (550 nm)[h] | 0.03, 0.20, 1.5 |
| Solar zenith angle (deg) | 0.0, 10, 20, 30, 40, 50, 60, 70, 75, 80, 85, 90[i] |

[a] Additional calculations with 200 DU and 400 DU for selected altitudes of 1 km and 10 km. [b] libRadtran default aerosol properties (Shettle, 1989). [c] Additional clear-sky calculations for 1 km ground elevation. [d] Mean ground albedo for vegetation (Feister and Grewe, 1995). [e] Cloud cases according to Table 2. [f] In-cloud altitude for a specific cloud case. [g] Spectral albedo scaled to produce ground albedos $A_{470}$ of 0.02, 0.04 or 0.07. [h] Default aerosol (AOD$_{550}$ = 0.23) scaled to produce aerosol optical depths AOD$_{550}$ of 0.03, 0.20 or 1.5. [i] 89.9°, the solar zenith angle range for calculations of spectral radiances with the solver DISORT is limited to <90°.

**Table 2.** Parameters of modeled cloud cases: cloud top and bottom heights, liquid water content (LWC) or ice water content (IWC), effective radii ($r_{\text{eff}}$), and cloud optical depths (COD).

| | top height (km) | bottom height (km) | LWC, IWC (g m$^{-3}$) | $r_{\text{eff}}$ ($\mu$m) | COD[a] |
|---|---|---|---|---|---|
| Clear-sky (Cl) | – | – | 0 | – | 0 |
| Cirrostratus (Cs) | 12 | 10 | 0.006 (ice) | 20 | 1 |
| Altostratus (As) | 3.7 | 3.3 | 0.29 | 7 | 25 |
| Stratus (St) | 0.2 | 0.0 | 0.58 | 7 | 25 |

[a] Approximate values for the cloud cases.

Five ground albedo cases were considered: (i–iii) a wavelength dependent ground albedo $A$ typical for vegetated ground, scaled to match values of 0.02, 0.04 and 0.07 at 470 nm, (iv) a high, wavelength independent, ground albedo of 0.8 representing snow cover, and (v) a spectral ground albedo of open water. The applied ground albedos are based on literature data. $A_{470}$ =



0.04 is considered a normal ground albedo. The theoretical case $A$=0 was included for test purposes but will not be used for the calculation of correction functions. More details on the ground albedos are given in Sect. S3.2 of the Supplement.

Three aerosol cases were implemented based on the default aerosol defined in libRadtran. The properties were varied by using the option to scale aerosol optical depth (AOD) to user-defined values at selected wavelengths, in this case at 550 nm. AODs for other wavelengths were scaled accordingly resulting in the following aerosol cases: (i) $AOD_{550} = 0.03$, (ii) $AOD_{550}$ = 0.2 and (iii) $AOD_{550} = 1.5$. These cases cover typical atmospheric properties from very clean oceanic to strongly polluted urban continental conditions. $AOD_{550} = 0.2$ is regarded as the normal aerosol optical depth. The theoretical case AOD=0 was
included but will also not be used to calculate correction functions. More details on the aerosol optical depth are given in Sect. S3.3 of the Supplement.

An overview of scenarios used for the platforms HALO and Zeppelin, as well as for the ground station is given in Tab. S1 of the Supplement. Not all possible combinations of cloud, albedo and aerosol cases were implemented as atmospheric scenarios. For HALO cruise flight altitudes below 200 m are unrealistic. The 200 m cloud top height of the St layer was
therefore chosen so that HALO is always above this cloud type for which the influence of different ground albedos was not evaluated. For the Zeppelin the St cloud case was neglected because visual flight rules do not permit in-cloud flights. Rare cases where the Zeppelin could be flying above low-lying clouds or ground fog are reasonably represented by scenarios with a high, wavelength independent ground albedo of 0.8. Altitudes below 50 m where also not considered for the Zeppelin because of the ground-shading effect of the airship itself. For ground-based measurements all scenarios for an altitude of 0 km were
taken into account except the St cloud case because radiance distributions turned out to be sufficiently similar for St and As cloud cases at ground level. Multiple cloud layers were also not considered. Such conditions are supposed to be covered by in-cloud scenarios and combinations of Cs or As cloud cases with a high ground albedo of 0.8.

Examples of modelled diffuse radiance distributions $L_\lambda(\vartheta,\varphi)$ for the upper and lower hemisphere under clear-sky conditions are shown in Fig. 7 for an altitude of 5 km, a solar zenith angle of 40° and a wavelength of 400 nm. In this example, the relative
contributions of direct, diffuse downward and diffuse upward radiation to the total spectral actinic flux density are 0.52, 0.26 and 0.22, respectively. For the same scenario, Fig. 8 shows azimuthal averaged spectral radiances for different wavelengths, normalized to their maximum values for better comparability. In both hemispheres these radiances are strongly enhanced at polar angles close to the horizon, except for 300 nm where the downward radiances are almost independent of polar angles.

With regard to Figs. 7 and 8 it should be noted that for the modelled spectral radiances polar angles $\vartheta$ were re-defined as
angles of incidence with respect to the $4\pi$ aircraft assemblies in accordance with the notations in the last sections. For the physical directions of propagation different polar angles ($\theta$) apply: $\theta = 180° - \vartheta$. The same holds for solar zenith angles, e.g. when the sun is located in the zenith ($\vartheta = 0°$) the radiation is directed towards the nadir ($\theta = 180°$). The use of angles-of-incidences has no consequences except that polar angle integration limits interchange for the upper and the lower hemisphere in some of the equations given in the following section 5.1.

Plots like those in Fig. 7 and 8 were produced for each atmospheric scenario, altitude, solar zenith angle and selected wavelengths. They provide a quick overview on the variation of radiance distributions and actinic flux densities as a function of atmospheric conditions. In Figs. S17 and S18 of the Supplement a second example is shown for the As cloud case under





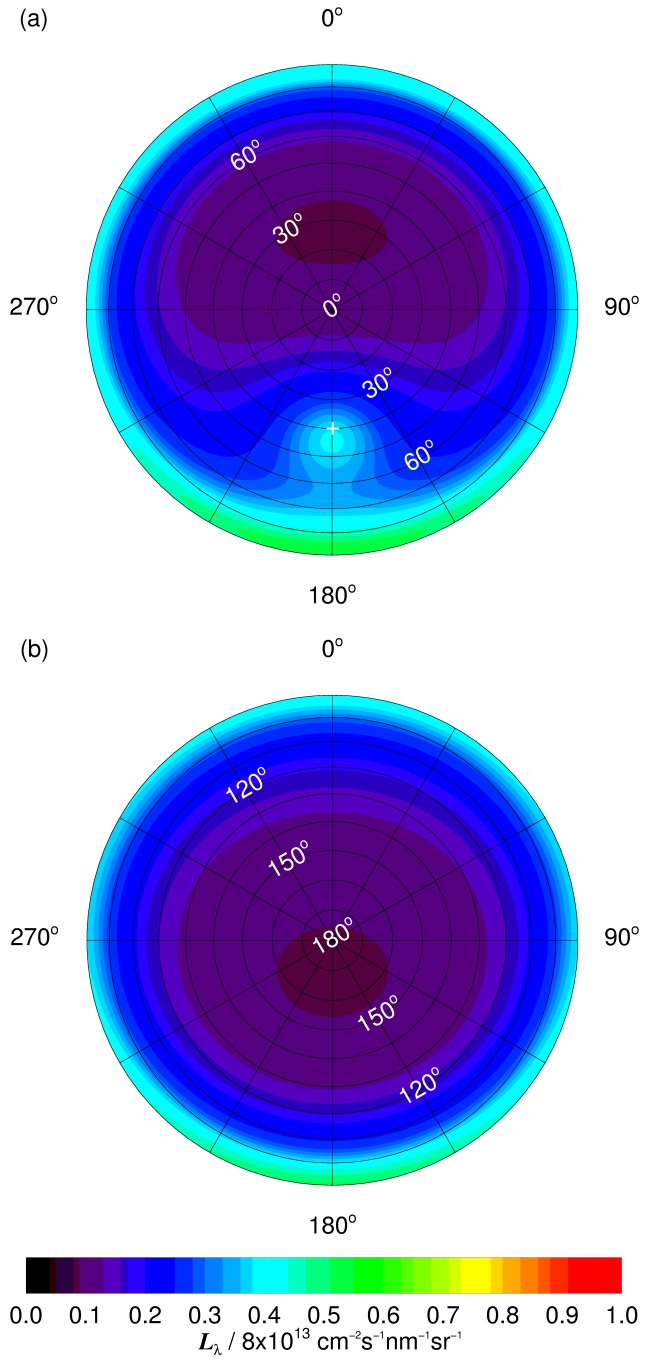

**Figure 7.** Contour plots of modelled diffuse spectral radiance distributions for a wavelength of 400 nm at an altitude of 5 km under clear-sky conditions at solar zenith and azimuth angles of 40° and 180°, respectively. (a) Downward spectral radiance. (b) Upward spectral radiance. Polar angles (white) are defined as angles of incidence. The position of the sun is indicated by the white cross in panel (a). In this example, ground albedos were scaled to 0.04 at 470 nm and aerosol optical depths to 0.2 at 550 nm (normal conditions). The colour scale was chosen for better comparability with Fig. S17 where the effects of an underlying As cloud layer are shown.



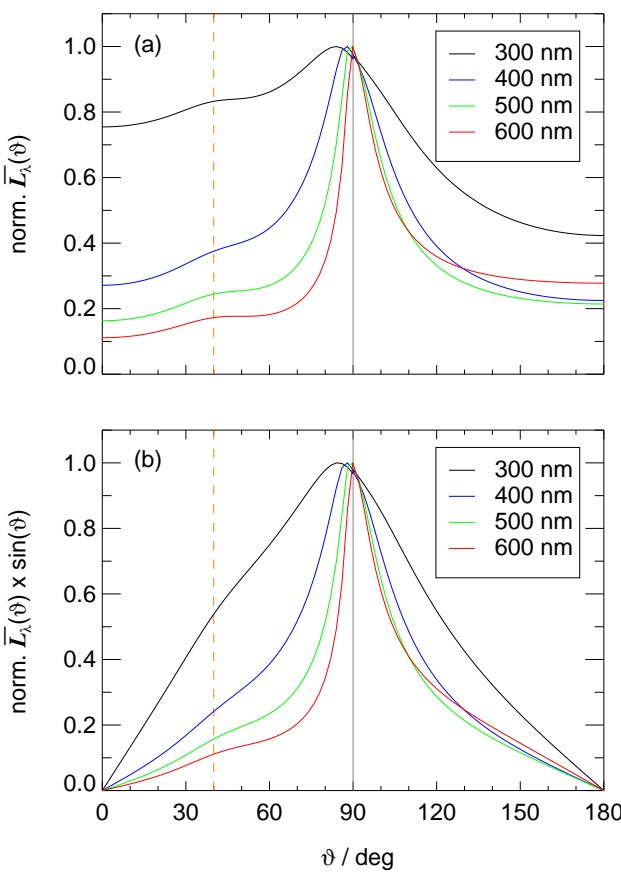

**Figure 8.** (a) Polar angle of incidence dependence of normalized, azimuthal mean diffuse spectral radiances for different wavelengths under the conditions in Fig. 7. (b) Azimuthal mean spectral radiances as in panel (a) but weighted with $\sin(\vartheta)$. The vertical grey line indicates the horizon, the dashed orange line the solar zenith angle. Direct sun contribution to spectral actinic flux densities for this scenario are 0.35 (300 nm), 0.53 (400 nm), 0.67 (500 nm) and 0.73 (600 nm). Compare with Fig. S18 of the Supplement.

otherwise the same conditions as in Fig. 7. Expectedly, the spectral actinic flux densities above the cloud layer are strongly enhanced by a factor of around 1.7 and the distributions are different for both upward and (to a minor extend) downward
spectral radiances. Two further examples of radiance distributions at a lower altitude under clear-sky conditions and below the As cloud layer are shown in Figs. S19-S22 of the Supplement. All model results are available for download for other users (Bohn, 2022). More details are given in Sect. S3.5 of the Supplement. The large number of model results naturally contains a lot of interesting information and phenomenons. However, a more detailed analysis is beyond the scope of this work. Potential uncertainties of the model results were also not considered. Rather the variability of naturally occurring radiance distributions
is assumed to be represented realistically by the different atmospheric scenarios.

For solar zenith angles approaching 90°, modelled spectral radiances will become unrealistic because diffuse radiation was calculated in plane-parallel geometry while for direct radiation a pseudo-spherical correction was applied in the model. On





the other hand, radiance distributions were found to change smoothly on a relative scale even at large solar zenith angles. Modeled radiance distributions for solar zenith angles of up to 85° are therefore considered useful but, except for ground-based

measurements, the correction procedure will anyway be limited to solar zenith angles smaller than 80° (Sect. 5.3.2).

## 5   Modelled correction functions

### 5.1   Definitions

Regardless of the more general definition given in Eq. 2, total solar spectral actinic flux density $F_\lambda$ can be separated into direct and diffuse components (e.g. Madronich (1987)):

$$F_\lambda = F_{\lambda,\text{dir}} + F_{\lambda,\text{dif}} = F_{\lambda,\text{dir}} + \int_0^{2\pi}\int_0^\pi L_\lambda(\vartheta,\varphi)\sin(\vartheta)\mathrm{d}\vartheta\mathrm{d}\varphi \tag{4}$$

For brevity the indication of the wavelength dependency of $F_\lambda$ and $L_\lambda$ variables was omitted here. Measurements can be simulated by calculating uncorrected spectral actinic flux densities $F_{\lambda,\text{m}}$ using the receiver assemblies' relative angular sensitivities $Z_\text{p}^\text{T}$:

$$F_{\lambda,\text{m}} = Z_\text{p}^\text{T}(\vartheta_\circ,\varphi_\circ)F_{\lambda,\text{dir}} + \int_0^{2\pi}\int_0^\pi Z_\text{p}^\text{T}(\vartheta,\varphi)L_\lambda(\vartheta,\varphi)\sin(\vartheta)\mathrm{d}\vartheta\mathrm{d}\varphi = Z_\text{S}^\text{T}\,F_\lambda \tag{5}$$

Angles are defined as angles of incidence and $\vartheta_\circ$ and $\varphi_\circ$ are corresponding solar zenith and azimuth angles, respectively (Sec. 4.2). Accordingly, the $Z_\text{p}^\text{T}$ have to be rotated horizontally to match the actual situation, dependent on the receiver heading and the solar azimuth angle. By analogy with the hemispherical correction function $Z_\text{H}$ introduced by Hofzumahaus et al. (1999), the right hand side of Eq. 5 defines a spherical correction function $Z_\text{S}^\text{T} = F_{\lambda,\text{m}}/F_\lambda$ for measured total spectral actinic flux densities. Because upward and downward $F_\lambda$ are determined separately and information on their contributions is relevant,

hemispherical corrections functions $Z_\text{H}$ are defined as well:

$$F_\lambda^\downarrow = F_{\lambda,\text{dir}} + F_{\lambda,\text{dif}}^\downarrow = F_{\lambda,\text{dir}} + \int_0^{2\pi}\int_0^{\pi/2} L_\lambda(\vartheta,\varphi)\sin(\vartheta)\mathrm{d}\vartheta\mathrm{d}\varphi \tag{6}$$

$$F_{\lambda,\text{m}}^\downarrow = Z_\text{p}^\text{Z}(\vartheta_\circ,\varphi_\circ)F_{\lambda,\text{dir}} + \int_0^{2\pi}\int_0^\pi Z_\text{p}^\text{Z}(\vartheta,\varphi)L_\lambda(\vartheta,\varphi)\sin(\vartheta)\mathrm{d}\vartheta\mathrm{d}\varphi = Z_\text{H}^\text{Z}\,F_\lambda^\downarrow \tag{7}$$

$$F_\lambda^\uparrow = F_{\lambda,\text{dif}}^\uparrow = \int_0^{2\pi}\int_{\pi/2}^\pi L_\lambda(\vartheta,\varphi)\sin(\vartheta)\mathrm{d}\vartheta\mathrm{d}\varphi \tag{8}$$

$$F_{\lambda,\text{m}}^\uparrow = Z_\text{p}^\text{N}(\vartheta_\circ,\varphi_\circ)F_{\lambda,\text{dir}} + \int_0^{2\pi}\int_0^\pi Z_\text{p}^\text{N}(\vartheta,\varphi)L_\lambda(\vartheta,\varphi)\sin(\vartheta)\mathrm{d}\vartheta\mathrm{d}\varphi = Z_\text{H}^\text{N}\,F_\lambda^\uparrow \tag{9}$$





Downward and upward $F_\lambda$ are indexed by downward and upward pointing arrows, respectively. The hemispherical correction functions $Z_{\mathrm{H}}^{\mathrm{Z}} = F_{\lambda,\mathrm{m}}^{\downarrow}/F_\lambda^{\downarrow}$ and $Z_{\mathrm{H}}^{\mathrm{N}} = F_{\lambda,\mathrm{m}}^{\uparrow}/F_\lambda^{\uparrow}$ refer to the zenith-oriented (Z) and nadir-oriented (N) top and bottom receivers on the upper and lower fuselage, respectively. Equations 6–9 apply to conditions $\vartheta_\circ \leq 90°$, i.e. no cases with upward direct radiation are considered but direct radiation unintentionally captured by the bottom receiver is included in Eq. 9.

An important constraint for the three correction functions is that total and hemispherical corrections are related to each other

dependent on the contributions of downward and upward actinic flux densities:

$$Z_{\mathrm{S}}^{\mathrm{T}} F_\lambda = Z_{\mathrm{H}}^{\mathrm{Z}} F_\lambda^{\downarrow} + Z_{\mathrm{H}}^{\mathrm{N}} F_\lambda^{\uparrow} \tag{10}$$

Any finally applied correction should comply with this equation to satisfy the general budget equation:

$$F_\lambda = F_\lambda^{\downarrow} + F_\lambda^{\uparrow} \tag{11}$$

For the special case of ground-based measurements of downward spectral actinic flux densities the integration limits can

be confined to the upper hemisphere if local upward radiation is negligible (low local ground albedo) or effectively shielded (extended artificial horizons):

$$F_{\lambda,\mathrm{m}}^{\downarrow\mathrm{G}} = Z_{\mathrm{p}}(\vartheta_\circ,\varphi_\circ)F_{\lambda,\mathrm{dir}} + \int\limits_0^{2\pi}\int\limits_0^{\pi/2} Z_{\mathrm{p}}(\vartheta,\varphi)L_\lambda(\vartheta,\varphi)\sin(\vartheta)\mathrm{d}\vartheta\mathrm{d}\varphi = Z_{\mathrm{H}}^{\mathrm{G}} F_\lambda^{\downarrow} \tag{12}$$

The corresponding correction functions were named $Z_{\mathrm{H}}^{\mathrm{G}} = F_{\lambda,\mathrm{m}}^{\downarrow\mathrm{G}}/F_\lambda^{\downarrow}$ and apply to the ground-station setup of the four receivers (Sect. 2), i.e. the $Z_{\mathrm{p}}$ in Eq. 12 correspond to those of the individual receivers (Fig. 4 and Fig. S4, Supplement). Other

ground-based applications will be discussed in Sect. 7.1.

### 5.2 Numerical calculations, uncertainties and exemplary results

The ground-station $Z_{\mathrm{H}}^{\mathrm{G}}$ of the four receivers and the three correction functions $Z_{\mathrm{S}}^{\mathrm{T}}$, $Z_{\mathrm{H}}^{\mathrm{Z}}$ and $Z_{\mathrm{H}}^{\mathrm{N}}$ for the airborne platforms were calculated for the atmospheric model scenarios and altitudes summarized in Tab. S1 of the Supplement. To avoid inaccuracies, numerical integrations were made after interpolating the variables to sufficiently high angular resolutions ($\leq 1°$). The

procedures were verified by comparing the numerically calculated $F_{\lambda,\mathrm{dif}}^{\downarrow}$ and $F_{\lambda,\mathrm{dif}}^{\uparrow}$ with the first-hand model output for these integrated quantities. The influence of different azimuthal positions of the sun was investigated by repeating the calculations after the spectral radiance distributions were rotated in $\varphi=2°$ steps until a full circle was accomplished, i.e. all possible receiver headings with respect to the sun were tested (180 calculations). Uncertainties for each calculation were obtained based on the uncertainty estimates of the $Z_{\mathrm{p}}$ variables (Sect. S2.1, Supplement) and of fuselage reflectivity, if applicable (Sect. S2.3,

Supplement).

### 5.2.1 Ground station

For the corrections on the ground, the results of the 180 calculations at different solar azimuth angles were averaged to obtain azimuthal mean $Z_{\mathrm{H}}^{\mathrm{G}}$ for downward spectral actinic flux densities. Averaging is justified because the azimuthal variations of the





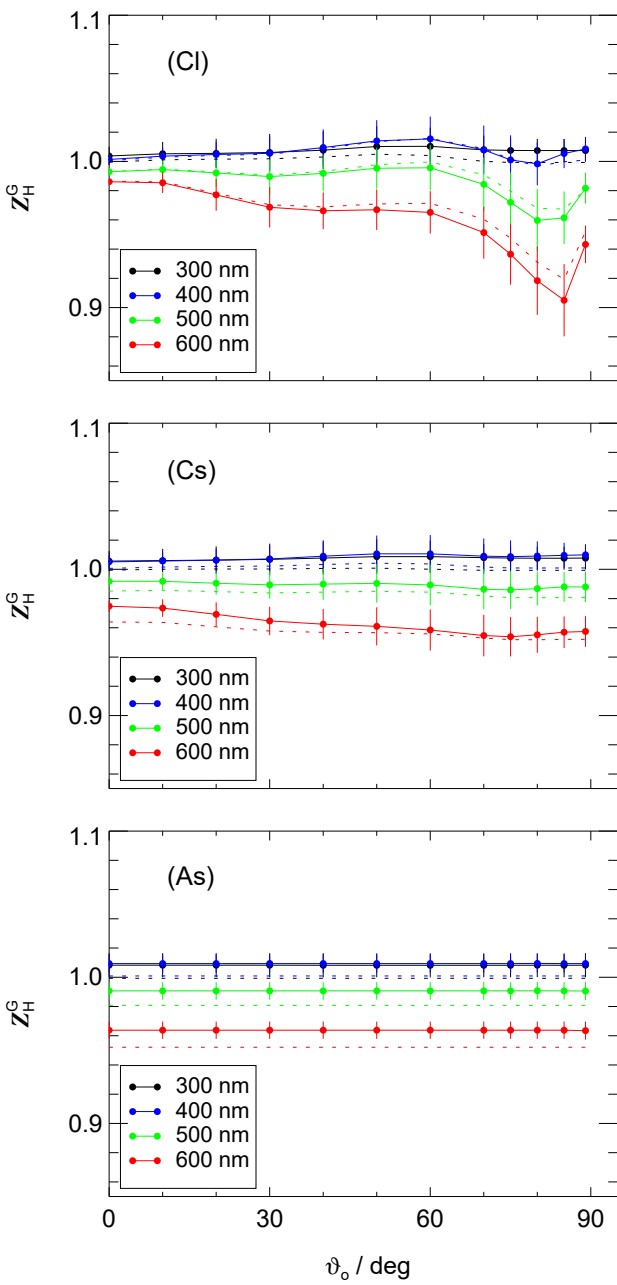

**Figure 9.** Modeled correction functions $Z_H^G$ for ground-based measurements of downward spectral actinic flux densities with the HALO top receiver as a function of solar zenith angle for selected wavelengths. Corrections apply to a scenario with normal aerosol load and ground albedos at different cloud cases. Upper panel (Cl): clear-sky; middel panel (Cs): Cs cloud layer; lower panel (As): As cloud layer. Dashed lines show results assuming isotropic distributions of downward diffuse spectral radiances for comparison.





$Z_\mathrm{p}$ variables are small (Figs. 3, 4 and Figs. S4, S5, Supplement). Total uncertainties for the averages were derived so that they

cover the uncertainties of the 180 calculations as well as the variations induced by the rotations of the radiance distributions.

As an example, Fig. 9 shows the resulting $Z_\mathrm{H}^\mathrm{G}$ for one of the HALO receivers on the ground for different cloud cases at normal aerosol optical depths and ground albedos. The solar zenith angle dependence and uncertainties are greatest under clear-sky conditions and smallest for the As cloud case where no direct radiation is present and the spectral radiance distributions exhibit no azimuthal dependencies. Overall, the corrections are small in the UV range ($\leq 2\%$) but can reach around 10% at 600 nm at

clear sky and low sun. The $Z_\mathrm{H}^\mathrm{G}$ for the other receivers under the same conditions are shown in Figs. S22-S24 of the Supplement. Expectedly, they are specific for each receiver, dependent on the individual angular sensitivities.

Considering other atmospheric scenarios, the influence of different ground albedos on the $Z_\mathrm{H}^\mathrm{G}$ was found to be minor ($\leq 1\%$) even for the greatest albedo of 0.8. On the other hand, the effects of aerosol load were more significant. The greatest AOD in the model led to clear-sky corrections similar to the Cs cloud case. Calculations for a ground elevation of 1 km instead of

sea-level produced minor deviations well below 1% even under clear sky conditions.

The dashed lines in Fig. 9 show corrections based on the assumption of isotropic diffuse radiance distributions in the upper hemisphere, i.e. only the contributions of direct and diffuse downward actinic flux densities were accounted for. The differences between dashed and full lines are small ($\leq 2\%$) which implies that for the determination of the $Z_\mathrm{H}^\mathrm{G}$ the use of modelled radiance distributions is expendable, at least for this receiver. The limited influence of the radiance distributions also means that the

correction functions remain applicable at solar zenith angles $>85°$ even though the radiative transfer model calculations are not be reliable under these conditions (Sect. 4.2).

### 5.2.2   Zeppelin

For the Zeppelin, the $Z_\mathrm{S}^\mathrm{T}$, $Z_\mathrm{H}^\mathrm{Z}$ and $Z_\mathrm{H}^\mathrm{N}$ were again averaged to obtain azimuthal mean values of the three correction functions. Azimuthal means are suitable because the azimuthal variabilities of the $Z_\mathrm{p}^\mathrm{T}$, $Z_\mathrm{p}^\mathrm{Z}$ and $Z_\mathrm{p}^\mathrm{N}$ variable distributions are small (Fig.

S11, Supplement). However, for the Zeppelin deviations from the normal flight attitude with zero pitch and roll angles can lead to additional variations in the corrections which increases the uncertainties. Attitude changes were specified by a tilt angle $\alpha$ with respect to the surface normal of the top receiver plane. A limit $\alpha = 5°$ was defined, after consulting tilt angle frequency distributions from the research flights where the $\alpha$ were calculated from airship pitch and roll angles. The limit $\alpha = 5°$ led to a typical loss in data coverage below 20% which is accepted to contain the uncertainties of the corrections and to ensure a

proper distinction of upward and downward actinic flux densities. To determine the influence of attitude changes, the azimuth dependent calculations (0–360°) were repeated eight times (with a resolution of 10°) after the $Z_\mathrm{p}$ variable distributions were tilted by 5° in eight directions with respect to the aircraft heading in 45° steps. Azimuthal variations expectedly increased upon a change in aircraft attitude, however strongly dependent on solar zenith angles and atmospheric conditions. The uncertainty estimates for the corrections were increased to cover the additional variations obtained for the eight tilted configurations.

As an example, Fig. 10 shows the altitude dependence of the $Z_\mathrm{S}^\mathrm{T}$, $Z_\mathrm{H}^\mathrm{Z}$ and $Z_\mathrm{H}^\mathrm{N}$ for the Zeppelin at normal aerosol optical depths and ground albedos for a solar zenith angle of 40°. The three lines of panels correspond to clear-sky conditions as well as overlying Cs and As cloud layers. The altitude dependencies of the $Z_\mathrm{S}^\mathrm{T}$ and $Z_\mathrm{H}^\mathrm{Z}$ are minor and insignificant for a





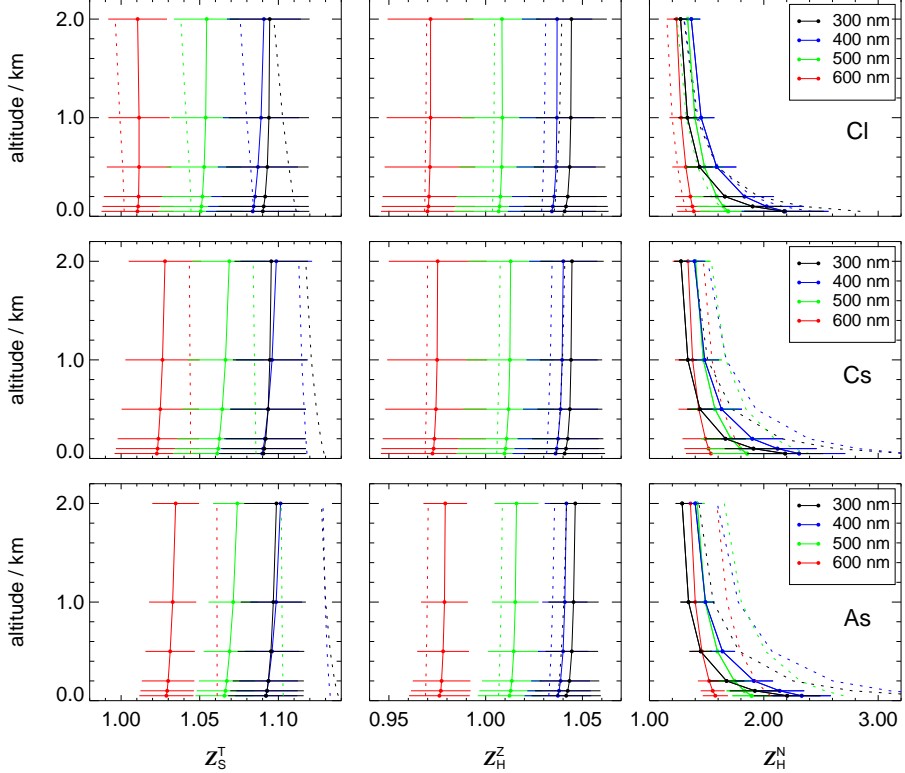

**Figure 10.** Altitude dependence of modelled Zeppelin correction functions $Z_S^T$, $Z_H^Z$ and $Z_H^N$ for total, downward and upward spectral actinic flux densities for a solar zenith angle of 40° and selected wavelengths. Corrections apply to normal aerosol load and ground albedos at different cloud cases. Top row (Cl): clear-sky; middle row (Cs): Cs cloud layer; bottom row (As): As cloud layer. Error bars include the effects of ±5° attitude variations. Dashed lines show results assuming hemispherical-isotropic distributions of downward and upward diffuse spectral radiances.

given cloud-case for all wavelengths within the estimated uncertainties which cover the effects of ±5° attitude variations as explained above. Because of insufficient field-of-view limitations of the bottom receiver, the $Z_H^N$ are generally greater
than unity. Moreover, they increase strongly towards the ground when upward actinic flux densities typically decrease which requires an increasing compensation of the cross-talk to the upper hemisphere. Accordingly, the increase towards the ground depends on ground albedos and virtually vanishes for the scenario with a high ground albedo of 0.8 (not shown). Generally, towards greater solar zenith angles uncertainty ranges increase with wavelength and decreasing aerosol optical depth for the clear-sky case but show little dependence on solar zenith angles for the cloud cases. Results for the same scenarios and a solar
zenith angle of 70° are shown in Fig. S26 of the Supplement.

Dashed lines in Fig. 10 correspond to corrections based on isotropic diffuse radiance distributions in each hemisphere using the modelled contributions of diffuse upward, diffuse downward and direct actinic flux densities. The differences are small for the clear-sky case, more pronounced for the $Z_S^T$ of the cloud cases and most significant for the $Z_H^N$ of the cloud cases where

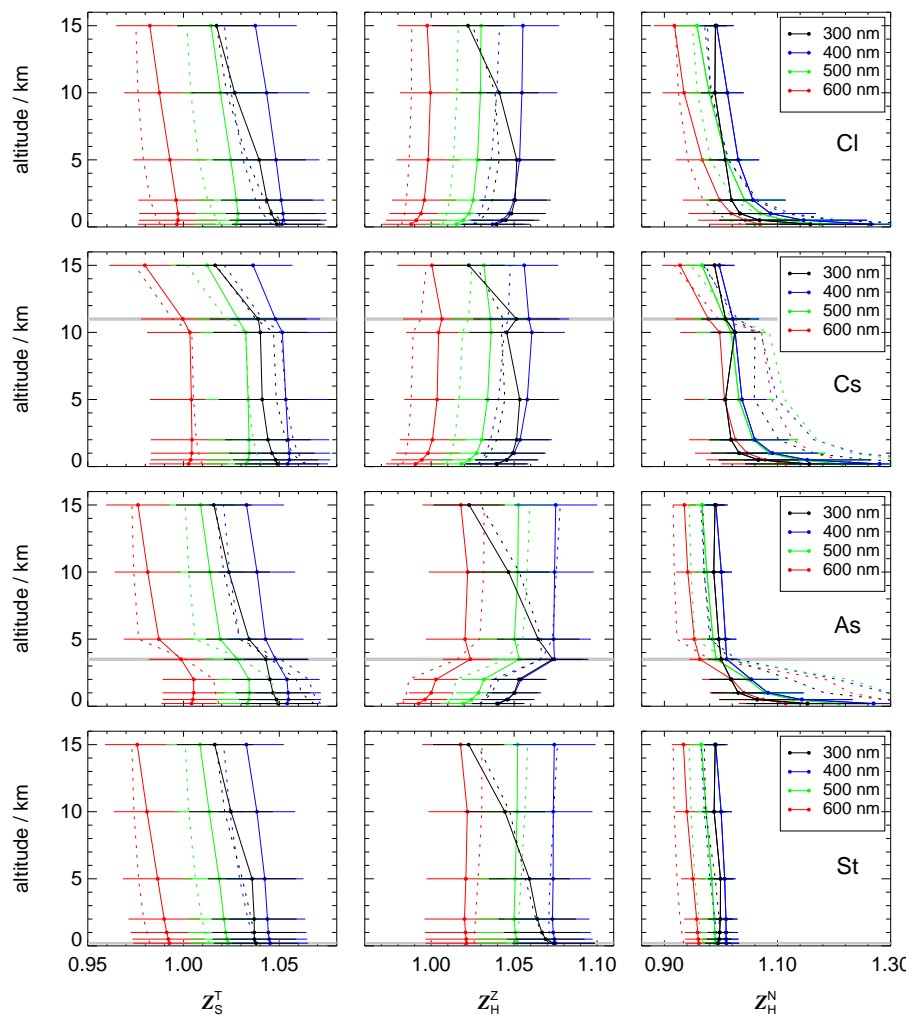

**Figure 11.** Altitude dependence of modelled HALO correction functions $Z_S^T$, $Z_H^Z$ and $Z_H^N$ for total, downward and upward spectral actinic flux densities for a solar zenith angle of 40° and selected wavelengths. Corrections apply to normal aerosol load, normal ground albedos and a solar heading angle $\gamma_\circ = 90\pm22°$ of the HALO configuration FLT for four cloud cases. Top row (Cl): clear-sky; middle upper row (Cs): Cs cloud layer; middle lower row (As): As cloud layer; lower row (St): St cloud layer. Cloud layer heights are indicated by horizontal grey lines. Error bars include the effects of $\pm2.5°$ attitude variations. Dashed lines show results assuming hemispherical-isotropic distributions of downward and upward diffuse spectral radiances.





the isotropic corrections are greater. This can be explained by the modelled downward spectral radiance distributions below
the cloud layers which show a decrease with increasing polar angle leading to a reduced cross-talk compared to the isotropic
case (Figs. S21, S22, Supplement).

### 5.2.3 HALO

For the three HALO configurations, simple azimuthal averages were not used because the $Z_\mathrm{p}^\mathrm{T}$, $Z_\mathrm{p}^\mathrm{Z}$ and $Z_\mathrm{p}^\mathrm{N}$ vary significantly
with azimuth angle at polar angles between around 80° and 100° (Fig. 5, Figs. S7, S9, Supplement). Consequently, the approach
described for the Zeppelin was refined for HALO. A solar heading angle ($\gamma_\mathrm{o}$) was defined describing the aircraft heading with
respect to the solar azimuthal position: $\gamma_\mathrm{o} = 0°$ when the aircraft was heading towards the sun and $\gamma_\mathrm{o} = 180°$ for the opposite
direction. Because the $Z_\mathrm{p}$ are similar on the left and right hand sides, the solar heading angle range was limited to 0–180°.
Correction functions were derived for solar heading angles of 0°, 45°, 90°, 135° and 180° by averaging the correction functions
obtained at $\alpha = 0$ within ±22° ranges of the five $\gamma_\mathrm{o}$ including results from left and right hand sides of the aircraft. Heading
specific uncertainties were determined from maximum deviations within the ±22° ranges including those obtained for the eight
tilted configurations. For HALO a more strict maximum tilt angle of $\alpha = 2.5°$ was defined because tilt angle distributions were
narrower compared to the Zeppelin. Nevertheless, corrections for $\alpha = 5°$ were also derived for HALO as a backup to optionally
increase data coverage at the expense of greater uncertainties.

Figure 11 shows the altitude dependence of $Z_\mathrm{S}^\mathrm{T}$, $Z_\mathrm{H}^\mathrm{Z}$ and $Z_\mathrm{H}^\mathrm{N}$ of the FLT configuration on HALO for different cloud cases
at a solar zenith angle of 40°. The results apply to a solar heading angle of 90°, i.e. with the sun on the left or right hand
side of the aircraft. The altitude range now expands up to 15 km and the fourth cloud case with the low-lying stratus layer is
included. In contrast to the Zeppelin, HALO can fly below and within clouds (Cs, As) as well as above all cloud types which
increases the ranges of modelled corrections. Towards the ground a similar, albeit less strong increase of the $Z_\mathrm{H}^\mathrm{N}$ was obtained.
This increase is smaller compared to the Zeppelin because the cross-talk to the upper hemisphere is, on average, smaller for
the HALO bottom receiver. For the St cloud case the increase of the $Z_\mathrm{H}^\mathrm{N}$ towards the ground is missing because upward actinic
flux densities are strongly enhanced. A comparable result was obtained for the maximum ground albedo of 0.8 (not shown)
which has a similar effect as the St cloud layer. Except below cloud layers, uncertainty ranges of the corrections, as well as the
dependence on solar heading angles and the HALO configuration generally increase with increasing solar zenith angles and
increasing wavelengths. Results for the same scenarios as in Fig. 11 for a solar zenith angle of 70° are shown in Fig. S27 of the
Supplement. The distinction of different solar heading angles helps to content the uncertainties of the corrections compared to
an approach using simple 360° azimuthal averages that were also derived.

Dashed lines again show the results assuming isotropic radiance distributions. The differences are less pronounced compared
to the Zeppelin but still significant for the $Z_\mathrm{H}^\mathrm{N}$ below cloud layers. On the other hand, under clear-sky and above cloud conditions
the assumption of isotropic radiance distributions in the lower hemisphere is apparently sufficient to obtain useful results.

A feature that stands out in Fig. 11 is the more pronounced altitude dependence of the $Z_\mathrm{H}^\mathrm{Z}$ for 300 nm. Modelled radiance
distributions vary significantly already within the narrow UV-B range (280–320 nm) dependent on total ozone columns. Nev-
ertheless, the influence of ozone columns on the corrections was found to be minor. At 1 km altitude, corrections obtained for



ozone columns of 200 DU and 400 DU are within 1% of the results for 300 DU for all solar zenith angles and wavelengths.
At 10 km altitude, deviations exceeding 1% were confined to solar zenith angels >80°. Consequently, the influence of ozone
columns was not considered in more detail. The validity of the finaly applied correction functions in the UV-B range for ozone
columns of 200 DU and 400 DU will be shown in Sect. 5.3.2.

## 5.3 Interpolations and parametrizations

### 5.3.1 Ground station

The dependence of the modelled $Z_H^G$ on atmospheric conditions is weak. Consequently, corrections for ground-based measure-
ments of downward spectral actinic flux densities can be calculated for each wavelength and solar zenith angle with uncertain-
ties covering all atmospheric scenarios including cloud cases and arbitrary azimuthal receiver orientations with respect to the
sun. The resulting uncertainties range around 2–3% in the UV range, dependent on receiver properties and solar zenith angles.
Final results for the four receivers examined in this work are shown in Fig. S28 of the Supplement for selected wavelengths.
Through interpolations these corrections become applicable to measurements under all conditions by interpolating corrections
and uncertainties as functions of solar zenith angles and wavelengths. Because of smooth changes with both variables these
interpolations introduce no additional uncertainties. In the UV-range, even constant $Z_H^G$ that are independent of solar zenith an-
gle and measurement conditions are sufficient. Further aspects and possible refinements related to ground-based measurements
are discussed in Sect. 7.1.

### 5.3.2 Airborne platforms

For the modelled $Z_S^T$, $Z_H^Z$ and $Z_H^N$ of the airborne platforms, corrections as a function of wavelength, solar zenith angle and
altitude alone are not useful because uncertainties become too large when all atmospheric scenarios are included, in particular
for the $Z_H^N$. Refinements by accessing measured aerosol loads or cloud presence are difficult because the required small-scale,
local information is usually not available along flight tracks. Moreover, the assignment to modelled scenarios is difficult in
particular for the cloud cases. Therefore parametrizations were developed which depend on the measured data alone and cover
all modelled atmospheric scenarios.

A closer look at the correction functions reveals that the most variable $Z_H^N$ increase strongly when the ratios of upward to
downward actinic flux densities go down, e.g. towards low altitudes at low ground albedos as explained in the previous section.
Therefore the ratio $\Phi_m$ of upward to downward uncorrected spectral actinic flux densities was used as a parametrization
variable. $\Phi_m$ has the advantage that it can be calculated directly from measured data in subsequent applications. For the
modelled corrections it is obtained from the following equation:

$$\Phi_m = \frac{F_{\lambda,m}^\uparrow}{F_{\lambda,m}^\downarrow} = \frac{F_\lambda^\uparrow Z_H^N}{F_\lambda^\downarrow Z_H^Z} \tag{13}$$

Plots of $Z_S^T$, $Z_H^Z$ and $Z_H^N$ as a function of $\Phi_m$ covering all atmospheric scenarios indeed show compact relationships for a given
altitude, solar zenith angle and wavelength. Examples for the Zeppelin at 1 km altitude are shown in Fig. 12. The $Z_S^T$ and $Z_H^Z$





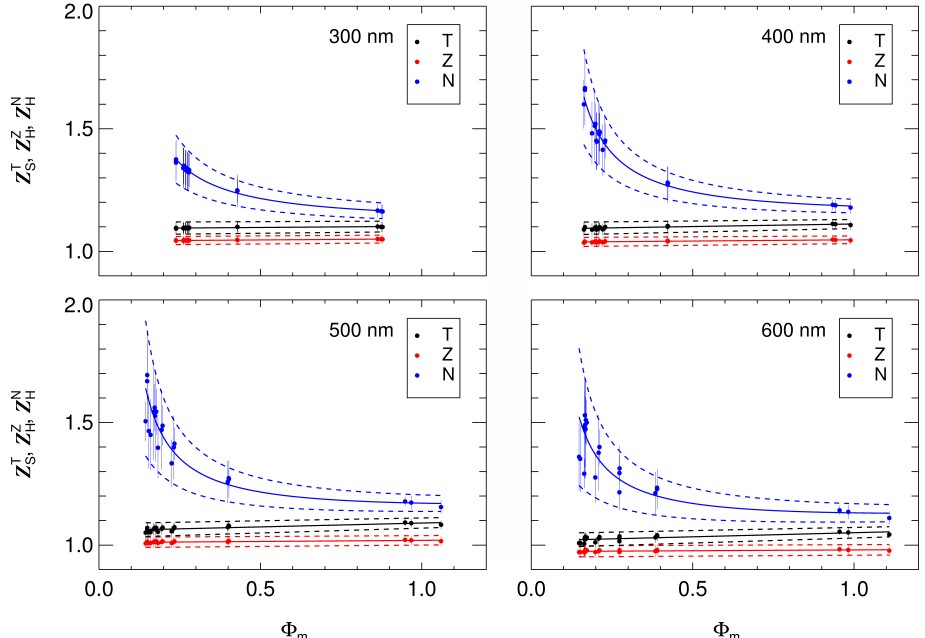

**Figure 12.** Correction functions $Z_S^T$ (T), $Z_H^Z$ (Z) and $Z_H^N$ (N) for the Zeppelin at an altitude of 1 km, a solar zenith angle of 40° and selected wavelengths as a function of $\Phi_m$ (ratio of uncorrected upward/downward spectral actinic flux densities). Data points with error bars show the results for all relevant atmospheric scenarios (Tab. S1, Supplement). Full lines are parametrizations with estimated uncertainty ranges indicated by the dashed lines. A second example for a solar zenith angle of 70° is shown in Fig. S29 of the Supplement.

are weakly dependent on $\Phi_m$ and can be described by linear dependencies in good approximations. Full black and red lines
show corresponding linear regressions. On the other hand, for the $Z_H^N$ linear approximations are inadequate in particular at lower altitudes and small $\Phi_m$. However, because the three correction functions are related to each other through Eq.10, the $Z_H^N$ that correspond to the linearly approximated $Z_S^T$ and $Z_H^Z$ can be calculated:

$$Z_H^N = \frac{Z_S^T \ Z_H^Z \ \Phi_m}{(1 + \Phi_m)Z_H^Z - Z_S^T} \tag{14}$$

Equation 14 ensures the consistency of the three corrections according to Eq. 10 and leads to an adequate description of the
observed non-linear dependence of $Z_H^N$ on $\Phi_m$ as shown by the full blue lines in Fig. 12. Ultimately, two linear parametrizations with four coefficients that depend on altitude, solar zenith angle and wavelength are required to describe the corrections within this approach for the Zeppelin.

Total uncertainties $\Delta Z_S^T$ and $\Delta Z_H^Z$ of the parametrized corrections were obtained by adding the deviations from the regression lines to the uncertainties of each scenario, followed by linear regressions of the uncertainties as a function of $\Phi_m$.
Corresponding upper and lower limits are indicated by the dashed black and red lines in Fig. 12. As the corrections themselves, the $\Delta Z_S^T$ and $\Delta Z_H^Z$ are weakly dependent on $\Phi_m$. On the other hand, the uncertainties of $Z_H^N$ are more variable and typically





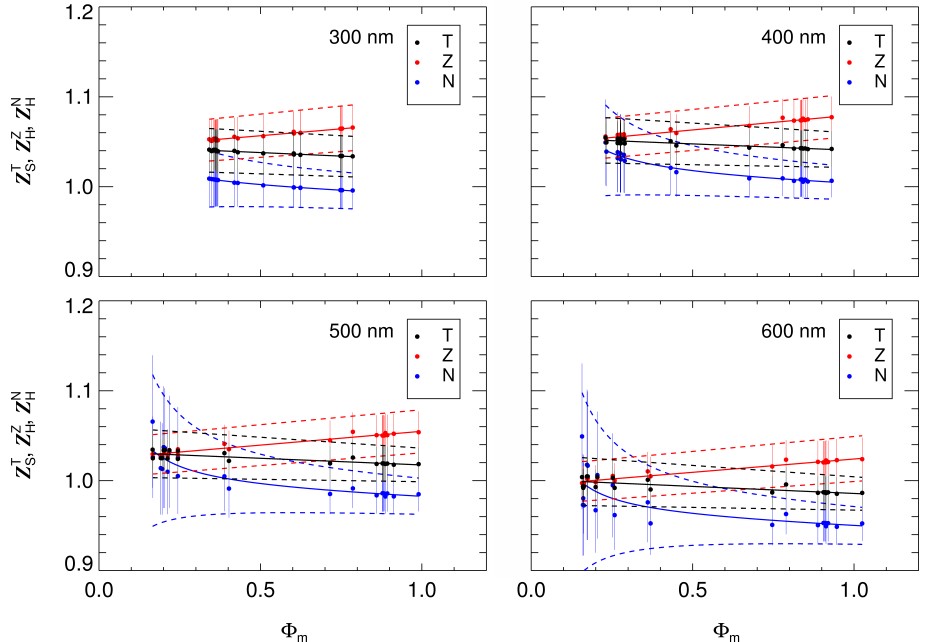

**Figure 13.** Correction functions $Z_S^T$ (T), $Z_H^Z$ (Z) and $Z_H^N$ (N) for HALO at an altitude of 5 km, a solar zenith angle of 40° and selected wavelengths as a function of $\Phi_m$ (ratio of uncorrected upward/downward spectral actinic flux densities). Data points with error bars show the results for all relevant atmospheric scenarios (Tab. S1, Supplement) for a solar heading angle $\gamma_o = 90°$ of the FLT configuration. Full lines are parametrizations with estimated uncertainty ranges indicated by the dashed lines. A second example for a solar zenith angle of 70° is shown in Fig. S30 of the Supplement.

increase non-linearly with decreasing $\Phi_m$. An adequate description was obtained by differentiating Eq. 14 with respect to $Z_S^T$ and $Z_H^Z$ to derive theoretical upper limits of $\Delta Z_H^N$ that were scaled by empirical factors $\delta_H^N$:

$$\Delta Z_H^N = \delta_H^N(\Phi_m) \times \left\{ \left| \frac{\partial Z_H^N}{\partial Z_S^T} \right| \Delta Z_S^T + \left| \frac{\partial Z_H^N}{\partial Z_H^Z} \right| \Delta Z_H^Z \right\} \tag{15}$$

The $\delta_H^N(\Phi_m)$ were again obtained from linear regressions as a function of $\Phi_m$ resulting in the upper and lower limits indicated by the dashed blue lines in Fig. 12. Typical values of $\delta_H^N$ range around 0.4 which is reasonable because the $\Delta Z_S^T$ and $\Delta Z_H^Z$ are not independent and partly compensate each other in the calculation of $Z_H^N$ (Eq. 14).

For the three different HALO configurations the same parametrization approach was used as for the Zeppelin, but separately for each of the the five solar heading angles. An example for an altitude of 5 km is shown in Fig. 13. At higher altitudes the

range of $\Phi_m$ generally becomes smaller and the non-linearity of the $Z_H^N$ is less pronounced. The distinction of different solar heading angles helps to contain the uncertainties, especially at large solar zenith angles and wavelengths.

Corrections from in-cloud model calculations at the intermediate altitudes of 3.5 km (As) and 11 km (Cs) were not considered in the parametrizations. Nevertheless, the in-cloud results are reasonably covered within the uncertainty limits of the





parametrizations using altitude-interpolated coefficients. Examples are shown in Fig. S31 of the Supplement. However, for the greatest model altitude of 15 km, no below-cloud scenario was included. This leads to decreasing uncertainties in the interpolation range between 10 km and 15 km which do not fully cover in-cloud or below cloud conditions at greater altitudes. On the other hand, the presence of clouds at flight altitudes >12 km was rare during previous research flights which justifies the current approach resulting in smaller uncertainties at very high altitudes.

Ozone columns other than 300 DU were also not included in the parametrizations. As was explained in the last section, the influence of ozone columns on the corrections was minor. A comparison of correction functions obtained at total ozone columns of 200 DU and 400 DU with the parametrizations derived for 300 DU is shown in Fig. S32 of the Supplement.

For both airborne platforms the overall performance of the parametrizations gradually degrade with increasing solar zenith angles and wavelengths resulting in correspondingly increasing uncertainties. At solar zenith angles >80° direct sun radiation can strike both receivers simultaneously which can result in strongly enhanced corrections dependent on wavelength and atmospheric conditions. Consequently, no corrections with acceptable uncertainty limits that cover all measurement conditions can be derived for solar zenith angles >80°. Exceptions are wavelengths below about 320 nm at all altitudes, as well as wavelengths below about 450 nm at low altitudes where the contributions of direct sun are sufficiently small. Anyway, for the present the application of the parametrizations is confined to solar zenith angles ≤80° which covers the predominant fractions of all research flights. Possible refinements for airborne measurements at solar zenith angles >80° will be discussed in Sect. 7. A detailed description of the correction procedure is given in Sect. S7 of the Supplement.

## 6 Applications to airborne measurements

### 6.1 Zeppelin flight example

An example of corrections derived for a Zeppelin flight under clear-sky conditions is shown in Fig. 14. On this day, the airship followed a quasi-stationary circular flight pattern for about four hours during which six height profiles were flown between about 100 m and 800 m above agricultural land in the Po valley, Italy, during the PEGASOS campaign (Li et al., 2014; Kaiser et al., 2015). The $Z_{\mathrm{H}}^{\mathrm{N}}$ show a wavelength dependent periodic pattern induced by the altitude changes. On the other hand, the $Z_{\mathrm{S}}^{\mathrm{T}}$ and $Z_{\mathrm{H}}^{\mathrm{Z}}$ and their uncertainties remain almost constant for a given wavelength within this flight's range of solar zenith angles.



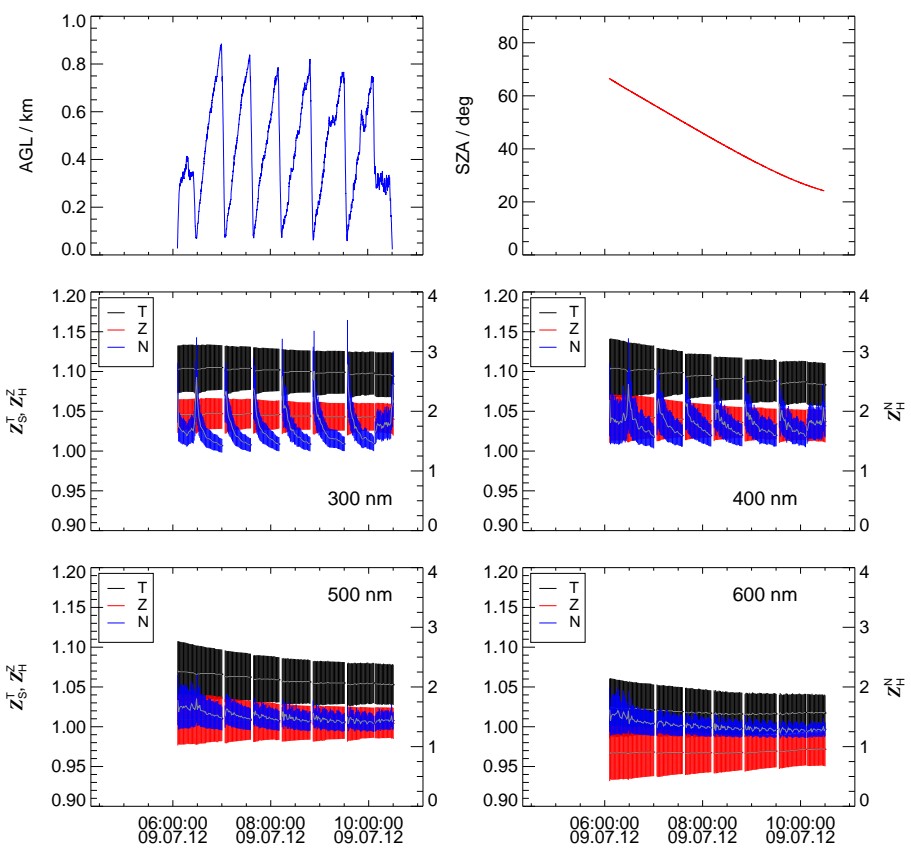

**Figure 14.** Zeppelin flight example with height profiles on 09 July 2012 about 40 km east of Bologna, Italy (PEGASOS campaign). Upper panels: Heights above ground, (AGL, left) and solar zenith angles (SZA, right) as a function of time of day (UTC). Middle and lower panels: Parametrized correction functions $Z_S^T$ (T), $Z_H^Z$ (Z) and $Z_H^N$ (N) for four selected wavelengths with error bars indicating uncertainties. For clarity, 1-minute averages are shown with grey overlays. The right hand $y$-axes refer to the $Z_H^N$.




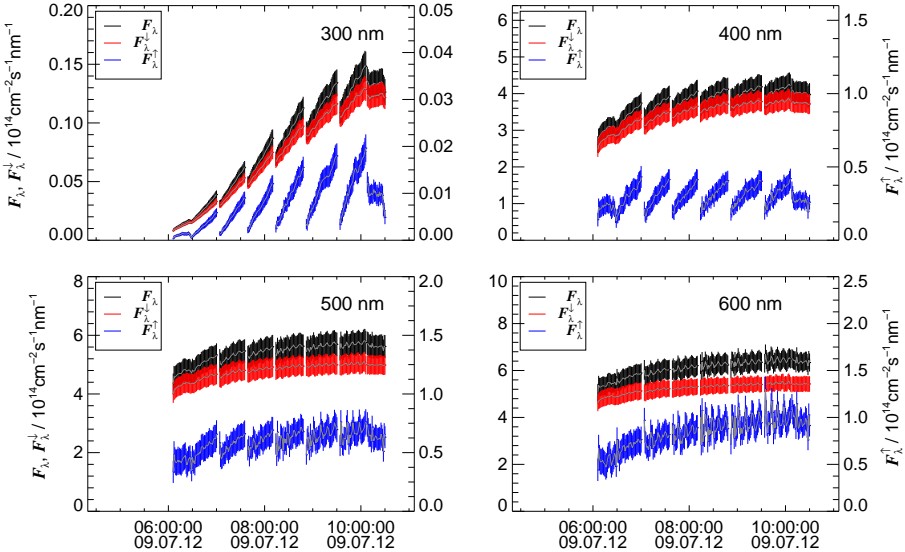

**Figure 15.** Total ($F_\lambda$), downward ($F_\lambda^\downarrow$) and upward ($F_\lambda^\uparrow$) spectral actinic flux densities of the Zeppelin flight shown in Fig. 14 for the four selected wavelengths. The error bars correspond to total uncertainties including those from corrections and calibrations (Sect. S7, Supplement). The right hand $y$-axes refer to the $F_\lambda^\uparrow$. The small, about 10-min periodic patterns were induced by the circular flight pattern.

The altitude dependence and the magnitude of the $Z_H^N$ decrease with wavelength which is explainable by increasing ground albedos over vegetated ground (Fig. S14, Supplement) and decreasing diffuse sky radiance in the upper hemisphere captured by the bottom receiver. However, despite values of around two for the $Z_H^N$ in the UV range, the $Z_S^T$ are merely increased by about 5% compared to the $Z_H^Z$ which is reasonable if only small fractions of the total actinic flux densities are directed upward.

The finally derived total, downward and upward spectral actinic flux densities are shown in Fig. 15 together with their total uncertainties. The different dependencies of the $F_\lambda$, $F_\lambda^\downarrow$ and $F_\lambda^\uparrow$ on altitude and solar zenith angle as a function of wavelength are qualitatively explainable. The increase of the $F_\lambda^\uparrow$ from 300 nm to 600 nm at the lowest altitudes is caused by the increasing ground albedos. On the other hand, the increase of the $F_\lambda^\uparrow$ with altitude is stronger for shorter wavelengths because of increased backscattering in the air column between the ground and the airship (Rayleigh and aerosol scattering). Increased scattering at shorter wavelengths also explains the different dependencies of the $F_\lambda$ on solar zenith angles. In addition, the influence of stratospheric ozone enhances the solar zenith angle dependence for 300 nm. Expectedly, photolysis frequencies show similar patterns dependent on the wavelength range of the photolysis reactions. However, a more detailed analysis of photolysis frequencies is beyond the scope of this study.

## 6.2 HALO flight example

For HALO flights, the spatial and atmospheric condition ranges were typically much greater than for the Zeppelin. An example is shown in Fig. 16 where HALO performed a nine-hour non-stop return flight from Taiwan to Japan over the East China





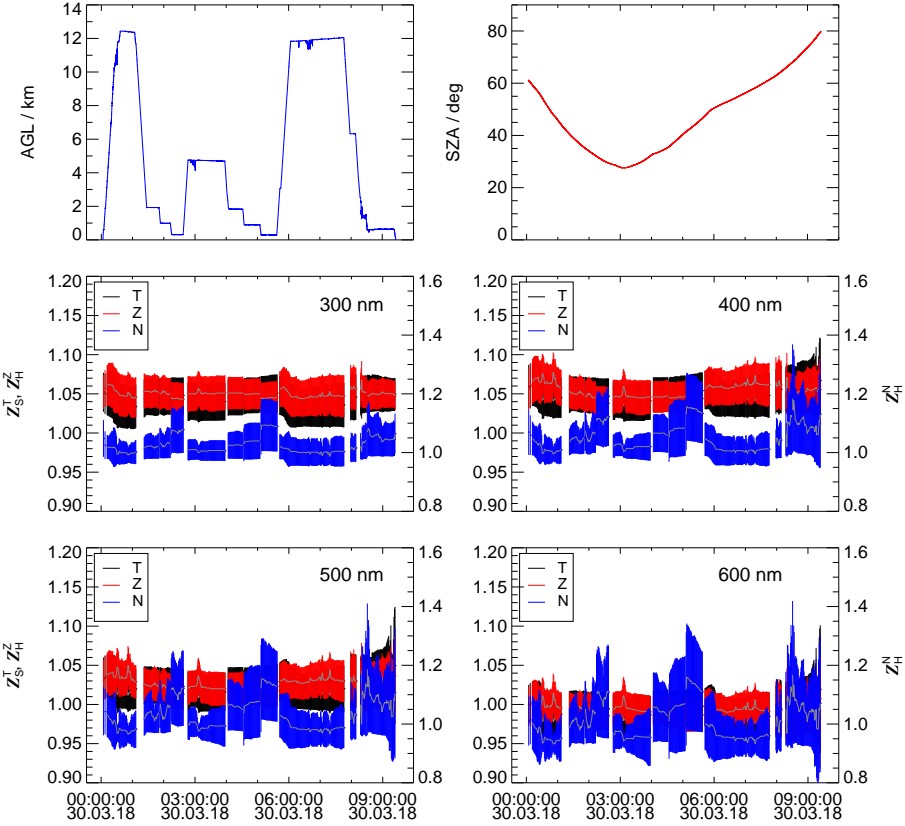

**Figure 16.** HALO flight example with a return flight from Taiwan to Japan on 30 March 2018 over the East China Sea (EMeRGe campaign). Upper panels: Heights above ground (AGL, left) and solar zenith angles (SZA, right) as a function of time of day(UTC). Middle and lower panels: Parameterized correction functions $Z_S^T$ (T), $Z_H^Z$ (Z) and $Z_H^N$ (N) for four selected wavelengths with error bars indicating uncertainties (FLT configuration). For clarity, 1-minute averages are shown with grey overlays. The right hand $y$-axes refer to the $Z_H^N$.

Sea during the EMeRGe-Asia campaign. Several flight levels between 0.5 km and 12 km were operated on this day under changing, partly cloudy atmospheric conditions. Again the $Z_H^N$ turned out to be most variable and uncertain, dependent on altitude and wavelength, but generally smaller compared to the Zeppelin. Minor, short term variations at constant altitudes indicate sporadic cloud influence. Gaps in the data record mark periods where flight manoeuvres led to attitude deviations that exceeded the HALO specific limit of 2.5°. Towards the end of the flight, solar zenith angles approached 80° resulting in increased uncertainties of the $Z_H^N$ at longer wavelengths.

The finally derived spectral actinic flux densities and their total uncertainties are shown in Fig. 17. They reveal a complex, dependence on altitude, solar zenith angle and cloud presence for the selected wavelengths. The variability of the $F_\lambda^\uparrow$ is strongly enhanced and values can become as high as the $F_\lambda^\downarrow$ through cloud influence. Accordingly, the total $F_\lambda$ increase during such periods. Cloud influence on $F_\lambda^\downarrow$ is hardly visible in this specific flight but clearly in others, in particular at low altitudes.





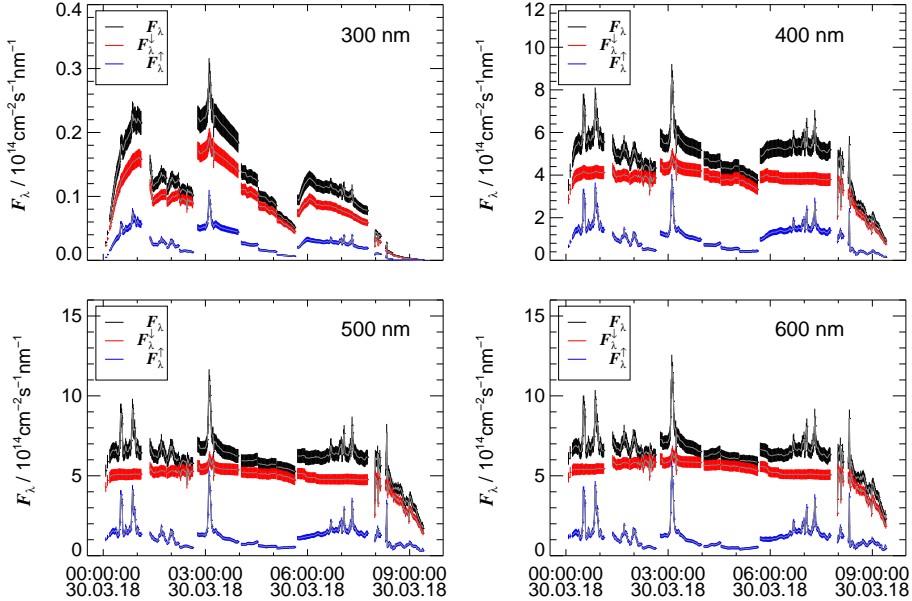

**Figure 17.** Total ($F_\lambda$), downward ($F_\lambda^\downarrow$) and upward ($F_\lambda^\uparrow$) spectral actinic flux densities of the HALO flight shown in Fig. 16 for the four selected wavelengths. The error bars correspond to total uncertainties including those from corrections and calibrations (Sect. S7, Supplement).

Because of wider ranges, the influence of altitude and solar zenith angles are greater than in the Zeppelin example. The
minor differences between 500 nm and 600 nm are explainable by similar scattering properties of air, aerosols and clouds as
well as similar ocean albedos. An analysis of these data with the help of radiative transfer model calculations is currently under
preparation but beyond the scope of this work. The corresponding photolysis frequencies again exhibit very similar, wavelength
dependent patterns. However, because of the greater altitude range, for some photolysis frequencies the additional influence of
temperature and pressure variations, affecting absorption cross sections and quantum yields, can become significant (Eq. 1).

## 7 Discussion

### 7.1 Ground based measurements

The correction functions $Z_H^G$ for measurements of downward spectral actinic flux densities are comparable with previous re-
sults for other receivers from the same manufacturer (Hofzumahaus et al., 1999; Bohn et al., 2008). Except for one receiver
and wavelengths >500 nm, the corrections remained below 10% with maximum uncertainties below 3%. Moreover, for the
four receivers used in this work, similar corrections were obtained using radiative transfer modelled and isotropic diffuse ra-
diance distributions in the upper hemisphere. This result probably also holds for other receivers with comparable properties
which simplifies the calculations. However, this does not mean that corrections for ground-based measurements are gener-
ally straightforward or secondary. Substantial corrections and large uncertainties can result for receivers with poorer reception



characteristics (Bohn et al., 2008) and, as already mentioned in the Introduction, the basically high accuracy of radiometric cal-
ibrations can be significantly degraded by uncertainties of receiver-related corrections. This issue may even remain unnoticed
unless the quality of receivers is thoroughly tested. On the other hand, as shown in Fig. S28, a constant correction factor cov-
ering all conditions can be sufficient in the UV-range. This is of relevance for measurements of $j(O^1D)$ and $j(NO_2)$ with filter
radiometers. If a calibration of these instruments is made by comparison with a corrected reference instrument, receiver related
mean corrections are already included (Bohn et al., 2004, 2008). In contrast, in the VIS range where significant contributions of
direct radiation are possible at large solar zenith angles, further refinements can be helpful. The potential presence or absence
of direct radiation increased the uncertainties of the $Z_H^G$ when all atmospheric scenarios were included (Fig. S28, Supplement).
Therefore, uncertainties can be reduced if conditions with and without direct radiation are distinguished, either based on the
measurements themselves, by the use of auxiliary instruments (sky cameras, pyrheliometers or sunshine recorders), or a sepa-
rate determination of the contribution of diffuse sky radiation. The latter is feasible using a classical shadow-ring, a sun-tracker
or a rotating shadow band (only one receiver required). Such approaches would for example be useful for a more accurate
determination of $j(NO_3)$ ($\lambda \leq 640$ nm) at low sun.

     A few more practical remarks for ground stations are added here. Naturally, for measurements of downward spectral actinic
flux densities the receiver should always be placed in a way that no other installations can cast shadows on it and that the
horizon is as possible unobstructed by nearby structures. Moreover, the cross-talk to the lower hemisphere should be minimized
by sufficiently large artificial horizons dependent on the local ground albedo as already noted by Hofzumahaus et al. (1999)
who estimated corrections of up to 15% for a ground albedo of 0.9 (fresh snow) with a 150 mm diameter artificial horizon. As
a consequence, the size of the artificial horizon (shadow ring) was doubled in subsequent applications of the same instrument
(e.g. Bais et al. (2003); Bohn et al. (2008)).

     Ground-based measurements of upward spectral actinic flux densities may be desirable as well, e.g. at sites with regular
snow cover. However, useful measurements of upward spectral actinic flux densities are challenging. First, downward facing
receivers capture the reflective properties of the natural or artificial ground at close range which may be different from the
ground in the surrounding area. A careful selection of the location is therefore important. For example, if measurements are
made on a pavement or artificial platform in an area dominated by vegetated ground, measured upward flux densities can be
misleading. There may be reasons why the local values are relevant but for most applications the influence of the surrounding
area is of greater significance. If no suitable location is available, an estimation of upward from measured downward flux
densities is possible based on typical ground albedos in the area (Madronich, 1987). Second, also a downward facing receiver
should be equipped with a large artificial horizon to prevent (i) cross-talk to the usually brighter, upper hemisphere and (ii)
reception of direct solar radiation at low sun, although this is a minor problem in the UV range as mentioned in Sect. 5.3. The
situation on the ground is comparable with the Zeppelin at very low altitudes where the limited size of the extension flange
produced overestimations by a factor of 2–3 in the UV range (Sect. 6.1). Similar overestimations are expected on the ground
(at low ground albedos) unless the upper hemisphere is effectively shielded. Of course, if required, a $4\pi$ correction approach
like for the Zeppelin can be implemented for a single, zero height above ground.



## 7.2 Airborne measurements

The correction functions $Z_{\mathrm{S}}^{\mathrm{T}}$, $Z_{\mathrm{H}}^{\mathrm{Z}}$ and $Z_{\mathrm{H}}^{\mathrm{N}}$ for the Zeppelin and HALO typically produce changes no greater than 5–10%.
An exception are the $Z_{\mathrm{H}}^{\mathrm{N}}$ at low altitudes and low ground albedos which can become significantly greater. The results are comparable with corrections applied in the literature for other airborne platforms. A direct comparison with previous work is difficult because the corrections are specific for each experimental setup and the individual receivers employed.

Volz-Thomas et al. (1996) used a prototype of the since then employed quartz dome receivers to measure $j(\mathrm{NO_2})$ with filter radiometers (370±40 nm) onboard a Lockheed C-130. The diameter of the base flanges was limited to 200 mm and the authors
optimized the total angular sensitivity with circular rims at the flange edges acting as artificial horizons. The performance of the $4\pi$ reception characteristics was tested in-flight by dedicated circular flight patterns with roll angles of 30° at different solar zenith angles which merely resulted in small variations of the total radiometer signals. From these test flights, uncertainties of the total $j(\mathrm{NO_2})$ caused by the $4\pi$ receiver imperfections of 1.5% and 6% were estimated for solar zenith angles below and above 75°, respectively. For downward and upward contributions under horizontal flight conditions, altitude dependent
correction factors in a range 1.00–1.04 and 0.69–1.01 were derived, respectively, with uncertainties of 2% and 5–12% at solar zenith angles ≤75°. These factors, which correspond to reciprocal values of the $Z_{\mathrm{H}}^{\mathrm{Z}}$ and $Z_{\mathrm{H}}^{\mathrm{N}}$ defined in this work, were derived based on radiative transfer calculations including the polar angle dependence of diffuse radiances, however confined to clear-sky conditions. In qualitative agreement with the results presented here, the corrections for the upward component increased with decreasing altitude leading to a minimum factor of 0.69 ($Z_{\mathrm{H}}^{\mathrm{N}}$=1.45) close to the ground.

Shetter and Müller (1999) employed a similar setup as Volz-Thomas et al. (1996) on a Douglas DC-8 for spectral actinic flux density measurements in a range 280–420 nm. No wavelength dependencies of angular sensitivities were detected and the effects of receiver imperfections were calculated assuming isotropic radiance distributions of diffuse sky radiation in both hemispheres. Average corrections of 1.036 and 1.027 which correspond to the $Z_{\mathrm{S}}^{\mathrm{T}}$ were finally derived for the UV-B and UV-A range, respectively, independent of measurement conditions with an estimated uncertainty of 4%. Because the work focussed
on photolysis frequencies from total spectral actinic flux densities, no separation of upward and downward components was made. In a follow-up study by Shetter et al. (2003) the DC-8 inlet configuration was modified and equipped with larger 300 mm artificial horizons (including rims) which resulted in close-to ideal angular responses in both hemispheres. Consequently, no corrections were applied for total, downward and upward spectral actinic flux densities and the remaining uncertainty was estimated 1.5%. The distinction of upward and downward contributions was confined to conditions where aircraft pitch or roll
angles did not exceed ±5°. A second, similar setup as on the DC-8 was installed on a Lockheed P-3B aircraft (Shetter et al., 2003; Lefer et al., 2003) and in-flight intercomparisons of the two instruments confirmed good agreements of $j(\mathrm{O^1D})$ and $j(\mathrm{NO_2})$ from total spectral actinic flux densities within 2% (Eisele et al., 2003).

Hofzumahaus et al. (2002) made clear-sky spectroradiometer measurements on a Falcon-20E aircraft in a range 280–420 nm. Similar to HALO, the smaller size of the aircraft did not allow for extended artificial horizons and the upward and downward
looking receivers were tilted in the flight direction by ±5° to compensate for the typical pitch angle. The overall angular sensitivity of the receiver assembly was comparable with that described in this work. The consequences of the non-ideal $4\pi$





behaviour were investigated by radiative transfer calculations including spectral radiance distributions under the measurement conditions. The deviations for total spectral actinic flux densities ranged from +1.4% (0.1 km) to +3.6% (12 km) at solar zenith angles <23° under clear-sky conditions. From these calculations a maximum 4% overestimation ($Z_S^T = 1.04$) was derived but no corrections were applied. Upward and downward components were not distinguished.

Jäkel et al. (2005) performed spectral actinic flux density measurements in a range 305–700 nm on a Partenavia P68-B in an altitude range below about 3 km. These authors used a stabilization system that kept the receivers horizontal within $\pm0.2°$ as long as pitch or roll angles did not exceed $\pm6°$. This system was originally designed for an accurate distinction of upward and downward spectral irradiances (Wendisch et al., 2001). The size of the artificial horizons was limited by the stabilization system to a diameter of about 130 mm. Consequently, the mutual cross-talk was significant and corrected for separately for the upward and downward looking receivers by adopting the concept of hemispherical correction functions using isotropic diffuse radiance distributions (Hofzumahaus et al., 1999). The wavelength and altitude dependence was investigated for clear-sky and cloudy conditions. For the downward component, a maximum correction of around 1.08 (= $Z_H^Z$) was obtained in the VIS range for an altitude of 2 km, above a highly reflective cloud. For the upward component, a maximum correction of around 1.35 (= $Z_H^N$) was reported in the UV-range for an altitude of 1 km under clear-sky conditions using a surface albedo of 0.08. The final corrections were made along the flight tracks by attributing measurement conditions to the modelled scenarios. The uncertainty of these corrections was estimated 2% .

Stark et al. (2007) made spectroradiometer measurements on a modified Lockheed WP-3 aircraft covering a wavelength range 280–690 nm. The setup followed that of Shetter et al. (2003) using a 300 mm artificial horizon with an outer rim. A correction function corresponding to the $Z_S^T$ was estimated for isotropic radiation, ranging between about 0.99 for 300 nm to 0.95 for 600 nm. These corrections were applied independent of measurement conditions which was accounted for by an additional 3% error. Upward and downward components were not distinguished.

Generally, on bigger aircraft, the base flanges that form artificial horizons can be larger without imposing aerodynamic issues. Under these circumstances, negligible corrections within small uncertainties can be achieved as demonstrated by Shetter et al. (2003). Moreover, a combination of two virtually ideal $2\pi$ receivers is expected to perform independent of aircraft attitude, as long as only total actinic flux densities are of interest (Shetter and Müller, 1999). On the other hand, even with two perfect, hemispheric receivers, a distinction of upward and downward flux densities requires close-to horizontal flight conditions or an active stabilization (Jäkel et al., 2005).

For HALO, the mutual cross-talk of the receivers and aircraft-specific field-of-view effects were more significant than in most previous studies which motivated the extended correction approach of this work. The effort is justified because of the large number of HALO flights for which corrections are required including further campaigns scheduled in the future. For the Zeppelin, mainly the cross-talk of the downward facing receiver to the upper hemisphere was significant and produced enhanced $Z_H^N$ under conditions with low ground albedo. The distinct dependence of the $Z_H^N$ on the parameter $\Phi_m$ was instructive to derive the parametrization concept which proved to be useful also for HALO. The main advantage of the parametrizations is that no potentially uncertain or unavailable information on the atmospheric state is required. Moreover, because different



wavelengths are treated separately, it is irrelevant whether or not the wavelength dependencies of ground albedos and aerosol optical depths in the model scenarios are realistic for the measurement conditions.

The use of isotropic radiance distributions for the calculation of the corrections led to slightly different results and cannot be recommended in general because the extent of the differences depends on receiver properties and atmospheric conditions. The

computational effort to derive the corrections is slightly lower but a wide range of conditions with different contributions of direct sun should be covered anyway. Moreover, under below-cloud conditions the assumption of isotropic radiances is clearly unrealistic for the upper hemisphere. Analytical expressions exist for the polar angle dependence of radiances under overcast conditions that can be easily implemented instead of constant radiances (e.g. Mayer and Kylling (2005)).

For the determination of total actinic flux densities and photolysis frequencies alone, the strict limitations with regard to

aircraft attitudes of 2.5° or 5° can be relaxed in order to increase data coverage. Uncertainties for total actinic flux densities could be determined for greater maximum attitudes, or alternatively, corrections and uncertainties could be calculated as a function of attitude. However, as is evident from the example flights shown in Figs. 15 and 17, the current attitude limitations are not critical for Zeppelin and HALO measurements.

The application of the parametrizations was limited to conditions with solar zenith angles ≤80° because corrections for

different atmospheric conditions become too variable when direct sunlight can strike both receivers simultaneously. This limitation affected a minor fraction of research flights on both HALO and the Zeppelin, but occasionally conditions with very low sun or day-to-night transitions were encountered. A reliable correction under such conditions would require an estimate of the contribution of direct sunlight (ideally based on the measurements themselves) and accurate radiative transfer model calculations at low sun including solar zenith angles >90°. As mentioned in Sect. 4.2, the currently applied radiative transfer

model in plane-parallel geometry will not give reliable results at low sun. The libRadtran package offers solutions in spherical geometry with advanced Monte Carlo solvers but these calculations are computationally more demanding. Moreover, a concept how to practically combine the model results with the measurements to derive useful corrections was not developed so far but may be worthwhile if twilight conditions become of greater interest e.g. for an accurate determination $j(NO_3)$.

## 8 Conclusions

Accurate measurements of spectral actinic flux densities require specific corrections to compensate for typical angular reception imperfections of optical receivers. A refined method to determine relative sensitivities of commonly used $2\pi$ solid angle optical receivers in the laboratory was presented in this work. The properties of four receivers were specified that were either employed separately on the ground to obtain downward spectral actinic flux densities, or pairwise on airborne platforms to measure upward and downward spectral actinic flux densities. Correction functions were calculated based on the relative sen-

sitivities, further platform characteristics (field-of-view effects, fuselage reflections) and spectral radiance distributions from a radiative transport model in a wavelength range $280 - 660$ nm for a number of atmospheric scenarios, intended to cover all realistic measurement conditions. The results were generally found to depend on wavelength and measurement conditions (solar zenith angle, altitude, ground albedo), including atmospheric variables (cloud cover, aerosol load). For ground-based measure-





ments, corrections for downward spectral actinic flux densities were determined and mean values as a function of wavelength
and solar zenith angle were derived with uncertainties covering all atmospheric scenarios. For airborne measurements, correc-
tions for upward, downward and total spectral actinic flux densities were calculated separately. Parametrizations of corrections
as a function of wavelength, solar zenith angle and altitude were developed that use upward/downward ratios of measured, un-
corrected actinic flux densities as input and provide uncertainties that cover all atmospheric scenarios. These parametrizations
reproduce the mutual dependence of corrections and their uncertainties resulting in consistent results for upward, downward
and (photochemically relevant) total spectral actinic flux densities. The application was limited to conditions with solar zenith
angles smaller than 80° and aircraft attitudes deviating less than 2.5° or 5.0° from normal flight conditions. Although all results
are receiver and platform specific, the method is generally applicable to other, comparable instruments and can improve the
accuracy of spectral actinic flux density measurements and resultant photolysis frequencies for many applications.

*Code and data availability.* libRadtran input file examples compatible with version 2.0.4 as well as spectral radiance output and corrections
for all atmospheric scenarios are available under https://doi.org/10.26165/JUELICH-DATA/8INBXK. Note that the corrections are specific
for the receivers and measurement configurations used in this work.

*Author contributions.* Both authors designed the study, made field campaign and laboratory measurements, as well as radiative transfer
calculations. BB made the final analysis and wrote the manuscript.

*Competing interests.* The authors declare that they have no conflict of interest.

*Acknowledgements.* The authors thank a great number of people who helped to get instruments airborne on the platforms Zeppelin NT and
HALO. We thank Enviscope GmbH for technical support with certifications and installations on HALO. We thank Zeppelin Luftschifftech-
nik and the Sensor and Data Group of DLR Flight Experiments department for providing aircraft avionic data. Regarding the campaign data
examples in this work, we particularly thank A. Kiendler-Scharr and T. F. Mentel (Forschungszentrum Jülich) for organizing the PEGASOS
Zeppelin campaign, as well as M. D. Andrés Hernández and J. P. Burrows (University of Bremen) for organizing the HALO EMeRGe
campaign. We thank the Japan Aerospace Exploration Agency for the provision of high-resolution ground-elevations from the ALOS Global
Digital Surface Model (AW3D30). We thank Bernhard Mayer, Arve Kylling and the other developers of the libRadtran model for making this
tool available for the scientific community. We thank Michael Decker (Forschungszentrum Jülich) for technical support with model calcula-
tions and the publication of the data. Finally, we thank the Deutsche Forschungsgemeinschaft (DFG) for funding under grants BO 1580/4-1
and BO 1580/5-1 within the HALO SPP 1294 priority program.



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
