# Peer review of "Optical receiver characterisations and corrections for ground-based and airborne measurements of spectral actinic flux densities"

_Atmospheric Measurement Techniques, 2022_

## Author Comment (AC1)

**Manuscript AMT-2022-288**

**Reply to Anonymous Referee #1**

In the following, referee comments are written in italics, our replies in normal font. Sections describing changes in the manuscript are indicated in blue. Figure and Table numbers refer to the original manuscript and supplement.

*Referee comment on "Optical receiver characterisations and corrections for ground-based and airborne measurements of spectral actinic flux densities" by Birger Bohn and Insa Lohse, Atmos. Meas. Tech. Discuss., https://doi.org/10.5194/amt-2022-288-RC1, 2022*

*This study examines actinic flux density optical collectors and develops a new technique to quantify correction functions for angular imperfections. The detailed responses of a pair of 2 pi optics used for a Zeppelin and a second pair used on the HALO aircraft. They are examined individually as downwelling ground based detectors and in 4 pi pairs as they perform on the aircraft. These optics show significant differences in the azimuthal and zenith responses and thus differ in the corresponding correction functions.*

*I commend the authors for the rigorous and exhaustive examination of the optical collectors and the deep analysis to correct imperfections on their specific measurements. The tools developed provide a resource to the community to improve actinic flux measurements. Importantly, these corrections do not rely on knowledge of the atmospheric environment (e.g. clouds, aerosols, etc) but rather on the relative changes in the measurements themselves.*

*This work provides a call to all groups with measurements of actinic flux to consider optical accuracy and the impacts of the non-ideal response. While this may not be practical for all groups, I suggest experts in the community may support more detailed optical analyses.*

*That said, the corrections are not always large and depend on measurement requirements. The authors note that the magnitude of corrections depends on the optical quality and measurement geometries. Downwelling optics for ground-based UV measurements (e.g. spectrometers or filter radiometers) may have relatively minor corrections (e.g. Figure S28), particularly with lower surface albedo. Total actinic flux from optical pairs with a sufficiently accurate 4 pi response also have relatively small corrections (e.g. Figure 11 and the discussion Section 7). The authors note that mechanical adjustment can improve hemispherical measurements, though corrections may still be necessary.*

*In addition, the primary purpose for such measurements is the subsequent calculation of photolysis frequencies. The impacts of the corrections on photochemistry are reduced and the authors note this is beyond the scope of this work. Nevertheless, it is an important consideration in the larger picture. Most photolysis frequencies are driven by UV wavelengths and the UV corrections in this study were relatively minor. In addition, photolysis frequency calculations include large molecular parameter uncertainties that typically exceed the measurement uncertainties. Nevertheless, better measurements are in fact, better, and molecular uncertainties may be reduced in future studies. At longer wavelengths, the uncertainties were more significant affecting the photochemistry of $NO_3$ and other molecules.*

*The paper is well written, relevant and fits well within the scope of AMT.*

Reply: We thank the referee for the positive evaluation of our work, the supporting general statements and helpful comments.

*Minor comments and revisions:*

*Line 1 and/or line 28: UV/VIS to "ultraviolet and visible (UV/VIS)"*

Reply: Was changed as recommended in both lines (in accordance with the journal rules).

*Lines 114-5: I do not see a description of FLT, FLV and FLN. Nor do I understand the acronyms. Figure S7 does give some information but does not distinguish between FLT and FLN.*

Reply: The acronyms were originally chosen to distinguish between ground and airborne configurations: GRD for "ground" and FLT for "flight". In a subsequent deployment the "nadir" receiver position was changed, and this configuration was named FLN. In another campaign both receivers were placed in the larger "viewport" adapters (Fig. S7) of HALO which led to the acronym FLV. We don't think the reader should be bothered with this acronym history. On the other hand, we noticed that the acronyms GRD and ZEPP_FLT were not mentioned in the manuscript but are used in some of the file names in the online material. We therefore added a sentence in the Supplement, page 14 line 27 to clarify: "The configurations can be inferred from the filenames: GRD refers to corrections for the four receivers on the ground, ZEPP_FLT to the Zeppelin configuration, and HALO_FLT, HALO_FLN and HALO_FLV refer to the three HALO configurations."
Moreover, we added a new figure (see below) after Fig. 2 showing the receiver positions of the three HALO configurations and added one sentence in line 117: "The receiver positions of the three configurations are indicated by arrows in Fig. X."

[Figure]

New figure X: "Top and bottom receiver positions of the three HALO configurations FLT, FLN and FLV. Adapted from a figure used with permission by DLR, Germany."

In addition, we reversed the flight direction of the setup in Fig. 2 to match with the new figure above.

[Figure]

Revised Fig. 2.

*Line 228: Typo. Change 'was' to 'were'*

Reply: Was changed.

*Figures 7, 8: The paper jumps between Zeppelin and HALO configurations. Be sure to distinguish the platform in each figure caption for clarity. In these two plots they are the Zeppelin optics.*

Reply: Figures 7 and 8 show radiance distributions which are independent of the platform (although the 5 km data were only used for HALO). We checked all figures to ensure that the respective platform is mentioned if necessary and, as a precaution, added the information more explicitly in the captions of Figs. 6, S9, S11 and S13.

*Line 358-60: Should this refer to Figures S4 and S5. Figure S11 does not show the azimuthal variability. Also, the variability of the nadir sensitivity looks to be about 5% based on the error bars in Figure S5. That does not seem to qualify as "small". Perhaps note it is an exception here.*

Reply: The reference Fig. S11 was wrong, we meant Fig. S12 and changed it. Regarding the azimuthal variabilities of the Zeppelin setup: these are small compared to those resulting for HALO at polar angles >80° (e.g. Fig. 5, Fig. S8, Fig. S10). Nevertheless, the sensitivities can vary in a 5% range for different azimuth angles for some receivers. But this variability is not neglected when azimuthal-averaged corrections are used. They are covered by the uncertainties (derived by the rotations of radiance distributions as explained in the text). However, we noticed that the azimuth-dependent $Z_P$ data of the four receivers was not included in the first version of the online material and added it.

*Figure S5: Just a comment: The bottom optic for the Zeppelin seems to be a significantly inferior optic. Perhaps that is why it was placed in the less consequential upwelling position. Could this be improved through mechanical adjustments? If so, the corrections would be more in line with the other optics. In addition, both Zeppelin optics seem to encourage crosstalk as they show 80% sensitivities at 90 degrees. Optimizing the angular response is a balancing act but perhaps these could be adjusted to be closer to 50%.*

Reply: Readjustment of the receiver optics is indeed difficult. We have tried this in the past with mixed success. We therefore rely on the adjustment skills of the manufacturer who hopefully achieved the optimum properties for each receiver. The around 80% sensitivity at 90 degrees is in general a good result and important to avoid large corrections on the ground, especially at low sun and long wavelengths. On the other hand, for the Zeppelin configuration the sensitivity peak of around 1.7 results at 90 degrees (Fig. S13). However, this peak would become very narrow and would hardly affect the corrections if the bottom receiver would perform like the top receiver. So, the key to optimum response is a high sensitivity up to 90 degrees plus an efficient horizontal shielding which was suboptimal for the Zeppelin bottom receiver. If the receivers were adjusted for a sensitivity of 50% at 90 degrees, this would result in a depression of sensitivity at smaller and larger polar angles which again would have to be compensated by corrections.

*Lines 370-380: I think this is an important place to note explicitly that the impact of the large upwelling correction near the surface has only a small impact on the total. Thus, the impact on total photolysis frequencies would be small.*

Reply: This is stated in line 500 where the Zeppelin measurement example is discussed. But it can be mentioned here already. We added one sentence in line 375: "However, the $Z_S^T$ are hardly affected by the greater $Z_H^N$ because the contributions of upward radiation are small under such conditions."

*Figures 11 and S27: The grey bars for the Cs cloud in these figures are deceiving. The cloud ranges from 10-12 km but is shown as a thin line at 11 km. I also suggest adding a point at 12 km to show the correction from the bottom to the top of the cloud.*

Reply: As suggested, we made additional simulations and derived corrections for the cirrus cloud case at 12 km. The figures were revised, and the cloud covers were included more realistically by grey areas indicating their depths (see revised Fig. 11 and Fig. S27 below). The 12 km corrections are close to those at 15 km and will not be used for the parameterizations (like the in-cloud corrections). In line 472 we added: "…at the intermediate altitudes of 3.5 km (As) and 11 km (Cs) as well as from above-cloud at 12 km (Cs) were not considered…" The 12 km data were included in version 2 of the online material.

[Figure]

Revised Fig. 11: "… Cloud layers are indicated by grey-shaded areas…."

[Figure]

Revised Fig. S27: "… Cloud layers are indicated by grey-shaded areas…."

Note that Fig. S27 will be removed from the Supplement following a comment by referee #2.

*Line 374: "Because of insufficient,…". This line is vague. If I am interpreting the meaning correctly, you could modify the end to, "Because of insufficient field-of-view limitations of the bottom receiver, significant cross talk to the upper hemisphere occurs and the Z(NH) are generally greater than unity".*

Reply: The sentence was changed as suggested.

*Line 410: Typo. I think you mean "contain the uncertainties"*

Reply: The referee is right, "content" was the wrong word. We changed to "confine the uncertainties" to clarify. We used a similar phrase in line 470 where we changed "helps to contain" to "again confines".

*Line 420: Typo. "final"*

Reply: That was a typo. We changed to "the finally applied".

*Line 504-5: The main source of the increase in upwelling flux from 300 to 600 nm is the orders of magnitude increase in the solar spectral shape. The albedo effect is secondary.*

Reply: Yes, this statement was wrong. We changed the sentence: "The increase of the $F_\lambda^\uparrow$ compared to the $F_\lambda^\downarrow$ from 300 nm to 600 nm at the lowest altitudes is caused by increasing ground albedos."

*Figure 17: The error bars show the uncertainties from the corrections and calibrations. I was disappointed not to see the impact of the corrections alone as that is the point of the entire paper. It should be shown in this figure (or another) and well explained in the text. The fact that they may be relatively small (especially on this HALO flight) is important.*

Reply: The uncertainties of the corrections are shown separately in Figs. 14 and 16 where they can be examined in greater detail. In Figs. 15 and 17 they are difficult to see for smaller values of the flux densities. Anyway, we now show both, total errors and those of the corrections alone as grey overlays. In the text we now write (line 502): "The finally derived total, downward and upward spectral actinic flux densities are shown in Fig. 15 together with their total uncertainties and those resulting from the corrections. The latter are dominant for the upward component and less significant for the total and downward." And in line 520 for HALO: "The finally derived spectral actinic flux densities and their uncertainties are shown in Fig. 17. The uncertainties from the corrections are again more significant for the $F_\lambda^\uparrow$ especially at low altitudes. Flux densities …"

[Figure]

Revised Fig. 15: …" The color-coded error bars correspond to total uncertainties including those from corrections and calibrations (Sect. S7, Supplement). The overlying grey error bars indicate the uncertainties from the corrections alone. …"

Note that the date was removed from the x-axis ("time (UTC)" was added instead), the x-range was changed, y-ranges for 400, 500 and 600 nm are the same now and panels are labelled (a)-(d) in response to referee #2.

[Figure]

Revised Fig. 17: "… The color-coded error bars correspond to total uncertainties including those from corrections and calibrations (Sect. S7, Supplement). The overlying grey error bars indicate the uncertainties from the corrections alone."

Note that the date was removed from the x-axis ("time (UTC)" was added instead), y-ranges for 400, 500 and 600 nm are the same now and panels are labelled (a)-(d) in response to referee #2.

*Section S3.4: Note that the relative contributions apply at 400 nm. That is stated in the figure captions but not the text.*

Reply: We added the information "for 400 nm" in the first sentence of the paragraph.

---

## Author Comment (AC2)

**Manuscript AMT-2022-288**

**Reply to Anonymous Referee #2**

In the following, referee comments are written in italics, our replies in normal font. Sections describing changes in the manuscript are indicated in blue. Figure and Table numbers refer to the original manuscript and supplement.

*The authors determined the angular response of optical inlets for measuring actinic flux densities and provided corresponding correction functions for ground-based and airborne applications and a wide range of atmospheric conditions. Their approach is based on detailed laboratory measurements combined with extensive radiative transfer calculations. Overall, the manuscript is well suited for publication in AMT. However, some comments might be considered before.*

Reply: We thank the referee for the positive feedback and helpful comments.

*General Comments*
*The authors give a lot of details to demonstrate what they did. The supplement material even extends their efforts to show everything that was made. This brings me to the main general comment. The reader might be overloaded by the number of plots and information specific to author's optics and applications. Therefore, the authors should consider limiting their plots to those that show significant differences. For example, it is not necessary to show the results for the four selected wavelengths for all investigations.*

Reply: We agree that a lot of material is presented. We tried to make each step as plausible and reproducible as possible with the help of the figures. The wavelength dependence in the measurement range is an important aspect of the study. The only figure (with extra panels) in the manuscript that shows rather small differences between different wavelengths is Fig. 17 (HALO measurement example, 400, 500 and 600 nm). However, it would be inconsequential to omit the wavelength dependence in this last plot showing the finally corrected data. We agree that in other figures in the Supplement the differences are small. To reduce the figure load, we removed Figs. S26, S27, S29 and S30 from the Supplement. The corresponding references were removed from the text in L379, L409, and the captions of Fig. 12 and Fig. 13.

*Specific Comments*
*P2L46: "the accuracy of measurements in the UV-B range" – I think it is rather a question of sensitivity what is meant here. Technically, the accuracy includes also the uncertainty due to a non-perfect angular response.*

Reply: What we mean is explained in the paragraph (i) below (L48-L54): it's a combination of enough UV sensitivity and a correct treatment of stray light which both affect the accuracy. We replaced "accuracy" by "specificity" in L46.

*P2L52: Maybe also refer to Jäkel et al. (2007) (https://doi.org/10.1175/JTECH1979.1) who discussed the stray light correction for a similar instrument.*

Reply: We were too much focussed on introducing our own accompanying paper. We changed the sentence to: "In previous studies suitable approaches were described for a widely used type of spectroradiometer (Jäkel et al., 2007; Bohn and Lohse, 2017)". The corresponding reference was added.

*P3L64: "…, owing to the greater importance of upward radiation, …" Why is it of greater importance?*

Reply: We extended the sentence: "…, owing to the greater importance of upward radiation, reflected by underlying air columns and clouds, …"

*P4L96: "of typically high spectral radiances in both hemispheres." This is not necessarily valid, e.g., flights performed over land or open water under cloud-less conditions show a low contribution of upward radiances.*

Reply: We discuss these cases in the succeeding sections. We changed "typically" to "commonly" which is certainly true for most airborne measurements.

*P7 Fig3: I'm wondering if this could be combined with the cross-section plots shown in Fig.4.*

Reply: Although the figures somehow belong to each other, they have different formats which makes it difficult to combine them without creating empty space in the resulting figure (which in addition would have to be downsized significantly). The figure sizes were chosen to fit in one column of the final two-column format of the journal. We are confident that the figures can finally be positioned next to each other.

*P8L172: "Because of the rotational symmetry of the receivers, dependencies on azimuth angles are minor." From the contour plot (Fig.3b), I would estimate a distinct variation of the angular sensitivity in azimuth direction (for 60° polar angle). Is this considered as minor dependence?*

Reply: Each receiver is different. As the example shows, there were some variations in a 5% range. These can still be considered as minor because they do not translate to a 5% variability of the measurements unless radiation is received from a single direction. We changed from "minor" to "typically minor (<5%)" to specify.

*P8L176: "The dependencies on polar angle and the wavelength Zp dependence are slightly different for the different receivers." Please give the range.*

Reply: We extended the sentence "… for different receivers but can differ by up to 15% at greater polar angles."

*P11 Fig5: Here, azimuthal averages are plotted. Think about to show also the corresponding standard deviation as in Fig. 4.*

Reply: The referee probably means Fig. 6. The standard deviations would be misleading here because the field-of-view effects can introduce strong azimuthal dependencies close to the horizon as shown in Fig. 3. Instead, we added the estimated uncertainties for the mean values and noted this in the caption of Fig. 4 and the corresponding Figs. S9, S11 and S13, see below. As mentioned in L195-199, Fig. 5 merely serves to visualize the (average) contributions of the top and bottom receivers. These mean data are not used to calculate corrections except for the hypothetical case of an isotropic radiance distribution as discussed in the paragraph L200-204. In this case the use of the mean values is justified. The final corrections are calculated based on the data shown in Fig. 5 (exemplarily for 400 nm). To clarify we extended the sentences in L204-206: "In order to obtain more realistic corrections, sensitivity distributions as shown in Fig. 5, as well as wavelength dependent direct sun contributions and diffuse spectral radiance distributions are required. The latter information is usually not available under measurement conditions."

[Figure]

Revised Fig. 6.: (a) Azimuthal averages of total relative angular sensitivities $Z_p^T$ (T) of HALO shown in Fig. 5 with contributions $Z_p^Z$ (Z) and $Z_p^N$ (N) of top and bottom receivers, respectively, for a wavelength of 400 nm (2°-interpolations). Error bars represent estimated mean uncertainties not covering azimuthal variabilities. The sensitivities of ideal 2π and 4π-receivers are shown for comparison (dashed lines). (b) The same data as in (a) but multiplied with sin($\vartheta$) to account for the $\vartheta$-dependence of solid angle contributions.

[Figure]

Revised Fig. S9.

[Figure]

Revised Fig. S11.

[Figure]

Revised Fig. S13.

*P12L200: "In panel (b) of Fig. 6 relative sensitivities were multiplied with sin(θ) to account for the solid angle contributions consistent with the θ-dependent areas in the projections of Figs. 3 and 5." I'm not sure if this step is obvious for the reader.*

Reply: The projections are introduced in L170. We extended the sentence to explain more clearly: "An azimuthal equal-area projection was chosen to correctly reproduce the solid angle contributions for different polar angles relevant for actinic flux density measurements, i.e. the areas increase with the sinus of the polar angle consistent with Eq. (2) (dω = sin(ϑ) dϑ dφ)."

*P13L242: "The applied ground albedos are based on literature data." Please give reference. Same for the cloud settings (L248).*

Reply: In L238 we added one sentence and three references related to the cloud microphysical properties: "These data represent typical values adopted from the literature (Miles et al., 2000; Sassen and Comstock, 2001; Krämer et al., 2009)." In L242 three references for the ground albedo were added: "(Bowker et al., 1985; Feister and Grewe, 1995; Wendisch et al., 2004)". The references are the same as already cited in the Supplement.

*P14L243: "… considered a normal ground albedo" Maybe it is better to call it a "default albedo" for your study. Same for the aerosol optical depth (L249).*

Reply: The term "default" is used several times already for the libRadtran default aerosol. With "normal" we meant to say not too high and not too low. We will replace "normal" by "standard" throughout the text which hopefully reflects the intended meaning.

*P14L263-P16L285: The authors present the simulations of the diffuse radiance field for cloud-less conditions. I would rather prefer to see a direct comparison to the more interesting cases that are shown in the supplement.*

Reply: As the referee noted, the number of figures in the manuscript is already large and we therefore shifted the other examples to the Supplement. A cloudless case is considered more common as a reference and is suitable to introduce the two representations of the radiance distributions in Figs. 7 and 8. The interested reader will certainly look for the other examples in the Supplement.

*P15 Fig.7: Could be combined with Fig.8.*

Reply: Here the same arguments hold as for Figs. 3 and 4 which we repeat here: The figures have different formats which makes it difficult to combine them without creating empty space. The figure sizes were chosen to fit in one column of the final two-column format of the journal. We trust that the figures will finally be positioned next to each other.

*P16 Fig.8: Is it reasonable to give azimuthal means here, since the distribution for the downward component has such a large azimuthal dependence? Think about a plot showing the principal plane direction instead.*

Reply: The azimuthal and polar angle dependencies for 400 nm are shown in the contour plot in Fig. 7 as an example, also for the principal plane. We think that the azimuthal averages in Fig. 8 are more relevant quantities for the integrating measurements examined here because radiation from a specific direction is never received exclusively. Moreover, if the azimuthal dependence of the receiver sensitivities were small, their means could be used directly together with the azimuthal means of the spectral radiances to derive the corrections for diffuse radiation (compare similarities of Figs. 6 and 8).

*P20L349: "clear-sky corrections": Does clear-sky corresponds with cloud-less conditions?*

Reply: Yes, throughout the text clear-sky refers to cloud-free conditions as is common in meteorology (cloud-cover=0). We stated this in line 234 and clarify here again by extending the sentence: "The greatest AOD in the model led to clear-sky corrections, i.e., corrections in the absence of clouds, like for the Cs cloud case."

*P22 Fig.11 / P23L390: "solar azimuthal position": Maybe use the term relative azimuth angle instead.*

Reply: We changed the sentence accordingly: "A solar heading angle ($\gamma_0$) was defined describing the relative azimuth angle of the aircraft heading with respect to the sun: …"

*P25 Fig.12: The number of dots should be 3 x 5 x 3 for this scenario, but it looks like less.*

Reply: The scenarios that were used for the analysis are listed in Tab. S1. However, the scenario groups T (3) and A (5) share one scenario (with standard ground albedo and aerosol) which was not explained. We therefore slightly revised the table caption: "The letter T (turbidity) denotes three scenarios with different aerosol optical depth cases at standard ground albedo ($A_{470} = 0.04$), the letter A (albedo) four additional scenarios with different ground albedo cases at standard aerosol optical depth ($AOD_{550} = 0.2$)." The total number of scenarios therefore is $(3+4) \times 3 = 21$ for the Zeppelin in Fig. 12 and $(3+4) \times 3 + 3$ (Str) = 24 for HALO in Fig. 13.

*P27L474: "altitude-interpolated coefficients": Do the atmospheric profiles of e.g. temperature and pressure have an effect on the altitude dependence?*

Reply: In terms of the corrections the effects are expected to be insignificant compared to those resulting for the different atmospheric scenarios. This is confirmed by the test calculations for a ground elevation of 1 km (900 mbar) which produced very small changes. As was mentioned in lines 282-285: "Potential uncertainties of the model results were also not considered. Rather the variability of naturally occurring radiance distributions is assumed to be represented realistically by the different atmospheric scenarios." In line 262, where the model input is described, we will add a statement to clarify: "Atmospheric pressure and temperature profiles were not varied. Their influence is presumed to be insignificant compared to that of the different atmospheric scenarios."

*P27L490: "A detailed description of the correction procedure is given in Sect. S7 of the Supplement." Maybe it is useful to provide a schematic that illustrates the correction procedure directly in the main manuscript.*

Reply: These technical details were deliberately moved to the Supplement. We produced a schematic (see below) which however cannot include all relevant details given in Sect. S7. We'll put the new figure next to Sect. S7 in the Supplement where the numbering of steps was consistently adapted.

[Figure]

New Figure X: "Schematic of data evaluation steps to derive corrections for airborne measurements. More details are given in Sect. S7. The final step of data selection (dependent on platform-specific selection criteria, e.g., minimum altitudes, shadings etc.), was omitted."

*P32L552ff: The paragraph partly contains advices which are obvious (e.g., "if measurements are made on a pavement or artificial platform in an area dominated by vegetated ground, measured upward flux densities can be misleading." In my opinion, it goes beyond the scope of this manuscript to go into the question of how to measure actinic radiation. Here the authors could shorten the text.*

Reply: We removed the two sentences following "For example, …" in line 570. Moreover, we removed the two sentences following "A few more practical remarks…." and start the paragraph with "Generally, for measurements of downward spectral actinic flux densities the cross-talk to the lower hemisphere should be minimized…"

*P35 Conclusions: I would suggest to give a final quantification of the corrections to illustrate their necessity.*

Reply: We included a sentence in L681: "The corrections derived in this work typically ranged well below 10% for total and downward spectral actinic flux densities but became more significant for upward spectral actinic flux densities dependent on the platform and atmospheric conditions."

*Technical Comments*

*P2L32: lifetimes – lifetime*

Reply: Was changed as suggested.

*P8L173: "are obviously invisible" - not visible ?*

Reply: We changed to "not visible".

*P9 Fig.4, P11 Fig.6: Please revise the legend. The dashed line is not shown as dashed line there.*

Reply: The dashed line appeared as a single, shorter line in the legend which may be confusing. We changed the line-style to make the legend clearer. As the same line-style was used in several figures, all were changed accordingly, i.e. in Figs. 6, S9, S11, S13 shown above and Figs. 4, S5 (not shown).

*P9l192: "in panel (a)" - give also figure number*

Reply: Was changed.

There are several figures without labeling the four panels.

Reply: We now consistently labelled different panels in all figures with (a), (b)… in Figs. 9-17. (Figs. 14-17 are shown below).

*P28 Fig.14: Maybe adjust the range of the x-axis (zoom in, e.g. 06:00 – 10:15 UTC). The date is not required as x-axis label.*

Reply: We changed the range of the x-axis, removed the date from the label and replaced it by "time (UTC)" to create a proper label, and rearranged the legends and labelling. Fig. 16 (HALO) was changed accordingly.

[Figure]

Revised Fig. 14.

*P29 Fig. 15: Use the same y-scale if appropriate. It helps to make the plots more comparable.*

Reply: The y-range of the three VIS-range panels are the same now. Labelling was made as in Fig. 14. Fig. 17 was changed accordingly.

[Figure]

Revised Fig. 15. Error bars from the corrections alone are shown in addition, in response to referee #1.

[Figure]

Revised Fig. 16.

Revised Fig. 17. Error bars from the corrections alone are shown in addition, in response to referee #1.